# FANCJ promotes PARP1 activity during DNA replication that is essential in *BRCA1* deficient cells

Ke Cong [1,4], Nathan MacGilvary [1,4], Silviana Lee [1], Shannon G. MacLeod [2], Jennifer Calvo[1], Min Peng[1], Arne Nedergaard Kousholt[3], Tovah A. Day[2] & Sharon B. Cantor [1] ✉

The effectiveness of poly (ADP-ribose) polymerase inhibitors (PARPi) in creating single-stranded DNA gaps and inducing sensitivity requires the FANCJ DNA helicase. Yet, how FANCJ relates to PARP1 inhibition or trapping, which contribute to PARPi toxicity, remains unclear. Here, we find PARPi effectiveness hinges on S-phase PARP1 activity, which is reduced in FANCJ deficient cells as G-quadruplexes sequester PARP1 and MSH2. Additionally, loss of the FANCJ-MLH1 interaction diminishes PARP1 activity; however, depleting *MSH2* reinstates PARPi sensitivity and gaps. Indicating sequestered and trapped PARP1 are distinct, FANCJ loss increases PARPi resistance in cells susceptible to PARP1 trapping. However, with BRCA1 deficiency, the loss of FANCJ mirrors PARP1 loss or inhibition, with the detrimental commonality being loss of S-phase PARP1 activity. These insights underline the crucial role of PARP1 activity during DNA replication in BRCA1 deficient cells and emphasize the importance of understanding drug mechanisms for enhancing therapeutic response.

Poly(ADP-ribose) polymerase inhibitors (PARPi) are effective in treating cancers with mutations in the hereditary breast and ovarian cancer genes, *BRCA1* or *BRCA2* (*BRCA*)[1,2]. The underlying PARPi sensitization mechanism remains unclear given that PARP1 has a range of distinct functions. The multifaceted roles of PARP1 span from excision repair, including single-strand break repair, base excision repair, and nucleotide excision repair, to mediating both classical and alternative non-homologous end joining, to the regulation of replication fork dynamics and DNA end resection activities[3–9]. Moreover, PARPi toxicity is thought to stem from chromatin-associated "trapped" PARP1. The inhibitors disrupt PARP1 catalytic activity, which in turn reduces the auto-poly(ADP-ribosyl)ation (auto-PARylation) that releases PARP1 from chromatin[10,11]. Thus, the collision of replication forks with either unrepaired single-strand breaks or trapped PARP1 complexes has been proposed to induce DNA double-strand breaks that necessitate BRCA1 function in homologous recombination (HR)[1,2]. Additionally, PARP1

functions in DNA replication as a backup to canonical lagging strand synthesis[12]. Upon detection of unligated lagging strands, PARP1 creates poly(ADP-ribose) (PAR) covalent attachments on itself and recruits XRCC1 and LIG3 to promote ligation. Accordingly, disruption of lagging strand synthesis and the trapping of PARP1 at unligated lagging strands could underlie toxicity[13].

We have reported that PARPi-induced single-stranded DNA (ssDNA) gaps are closely aligned with PARPi toxicity. Gaps as a determinant of PARPi response were in part highlighted by the comparison between cells deficient in the hereditary breast and ovarian cancer genes *BRCA1* or *FANCJ*. Cells deficient in either gene display similar defects in HR and fork protection[14–16]. By contrast, BRCA1 and FANCJ deficient cells differ in their relationship to PARP1. PARPi induced few replication gaps and little sensitivity in FANCJ-deficient cells as compared to high gap accumulation and synthetic lethality with BRCA deficiency. Moreover, PARP1 activity during S phase is abnormally low

[1]Department of Molecular, Cell and Cancer Biology, University of Massachusetts Chan Medical School, Worcester, MA 01605, USA. [2]Northeastern University Biology Department 360 Huntington Avenue, Mugar Life Science Building, Rm 220, Boston, MA 02115-5005, USA. [3]Division of Molecular Pathology, Oncode Institute, The Netherlands Cancer Institute, 1066CX Amsterdam, the Netherlands. [4]These authors contributed equally: Ke Cong, Nathan MacGilvary. ✉e-mail: sharon.cantor@umassmed.edu

in FANCJ-deficient cells but elevated in BRCA1 deficient cells. Finally, while replication speed is also distinct between FANCJ versus BRCA1 deficient cells, a series of cell models revealed that speed can be uncoupled from PARPi toxicity[17]. The contrast between BRCA1 and FANCJ is further supported by a genetic interaction network of PARPi response[18]. Notably, *FANCJ* loss as compared to other *BRCA*-Fanconi anemia genes confers the weakest PARPi sensitization phenotype, indicating again the unique role of FANCJ. Together, these findings suggest that PARP1 replication activity is central to the induction of PARPi-induced gaps and sensitivity, but how FANCJ confers these outcomes is unclear.

We hypothesized that *FANCJ* loss could impact PARP1 activation via changes in the replisome composition and/or DNA secondary structures. G-quadruplexes (G4s) form in lagging strands and are elevated in the replisome of FANCJ deficient cells[19–21]. G4 processing by FANCJ requires its helicase and translocase activities as well as the integrity of lysines 141 and 142 that bind G4s[22,23]. These lysine residues also mediate an interaction between FANCJ and the mismatch repair (MMR) protein MLH1 that is required for replication stress recovery. Restart in cells lacking the FANCJ-MLH1 interaction can be restored by depletion of the MMR protein MSH2, suggesting that dysregulated MMR limits replication in FANCJ deficient cells[24,25]. Intriguingly, beyond its role in correcting post-replication DNA mismatches, MMR has heightened activity on lagging strands[19–21]. Restricting MMR during replication could potentially be the mechanism by which FANCJ activates PARP1 backup function on the lagging strand. This may clarify why, in FANCJ deficient cells, there is a modest presence of PARP1 activity, PARPi-induced ssDNA gaps and sensitivity[17].

Here, we uncover that PARP1 function during DNA replication is disrupted in FANCJ-deficient cells even though PARP1 chromatin loading and activation by DNA damaging agents remain intact. Our data support a model in which FANCJ dismantles replisome-associated MSH2-bound G4s that limit ssDNA and PARP1 activation. This model is supported by the finding that MSH2 depletion re-establishes PARP1 activity along with PARPi-induced gaps and sensitivity in cells that lack the FANCJ-MLH1 interaction. Based on the discovery that BRCA1 deficient cells have elevated PARP1 activity and extreme sensitivity to any perturbation that disrupts PARP1 replication activity, we can further draw the conclusion that loss of PARP1 activity as opposed to canonical PARP1 trapping is the primary cause of cytotoxicity in these cells. Given that PARP1 trapping is toxic in other contexts, our findings underscore the importance of understanding drug action in precision medicine, to facilitate optimal drug combinations and prevent resistance that could evolve from low PARP1 S-phase activity and PARP1 trapping.

## Results

### FANCJ promotes PARP1 activity during DNA replication and limits MSH2 loading at the replisome

We previously found minimal PARPi sensitivity in FANCJ deficient cells including human retinal pigment epithelial 1 (RPE1), osteosarcoma (U2OS) and immortalized human embryonic kidney 293T cells[17]. Further characterization of *FANCJ* knockout (KO) RPE1 cells revealed aberrantly low PARP1 activity as measured by PARylation (PAR) in contrast to *BRCA1* KO cells in which PAR was higher than wild-type (WT) controls[17]. Thus, we sought to determine if PAR was also low in the *FANCJ* KO U2OS and 293T cell systems. To identify the endogenous PARP activity without external genotoxic stress as previously reported[12], we utilized an inhibitor of PARG, an enzyme primarily accountable for poly(ADP-ribose) removal, and observed elevated poly(ADP-ribose) or PARylation that correlated with replication[17]. Similar to the RPE1 cells, we observed low PAR in the *FANCJ* null U2OS cells as measured by immunoblot following PARGi incubation (Fig. 1a, b). Low PAR was also detected in *FANCJ* null 293T cells by immunoblot (Fig. 1c, d). Similar to our prior findings in RPE1 cells, the *FANCJ* KO U2OS or 293T cells activated PARP1 following treatment with

DNA damaging agents, methyl methanesulfonate (MMS), and hydrogen peroxide ($H_2O_2$) (Supplementary Fig. 1a, b)[17] demonstrating there was not an intrinsic defect in PARP1 enzyme activity and that *FANCJ* was not required for PARP1 activation in response to DNA damage. As found in RPE1 cells, we also observed that the lower PARP1 activity was unique to S phase or actively replicating U2OS cells positively incorporating the nucleotide analog 5-ethynyl-2′-deoxyuridine (EdU) as measured by immunofluorescence (Fig. 1e).

The low S-phase PARP1 activity in cells without the DNA helicase/translocase activity of FANCJ could result from either a failure to efficiently load or unload PARP1 from chromatin (Fig. 1f). Thus, we sought to address the impact of FANCJ on replication-associated PARP1. We first re-examined our prior iPOND (isolation of proteins on nascent DNA) data. As compared to WT, *FANCJ* KO 293T cells had enriched replisome and overall chromatin-associated PARP1[16]. Consistent with this finding, we observed greater chromatin bound (CB) PARP1 in the *FANCJ* null RPE1 cells that was selective to EdU positive cells and was only modestly increased following PARPi treatment (Fig. 1g, h and Supplementary Fig. 1c). The PARP1 replisome enrichment in *FANCJ* KO RPE1 cells was also observed by proximity ligation assay (PLA) compared to a PCNA-PCNA PLA control (Fig. 1i and Supplementary Fig. 1d). Together, these findings suggest that FANCJ is dispensable for PARP1 chromatin localization but rather critical for its activation and/or unloading (Fig. 1f).

To understand how FANCJ could promote PARP1 unloading and/or S-phase activation, we considered that in addition to PARP1, the MMR proteins MSH2 and MSH6, which form the MutSα complex, were elevated in the replisome of *FANCJ* KO 293T cells as suggested in our previous iPOND study[16]. We also found enriched MSH2-PCNA PLA signal and overall enriched CB-MSH2 in *FANCJ* KO RPE1 as compared to WT cells (Fig. 2a and Supplementary Fig. 2a). Furthermore, we observed a more pronounced MSH2-PARP1 PLA signal in *FANCJ* KO RPE1 cells (Fig. 2b) suggesting the proximity between PARP1 and MSH2 were aberrantly elevated and proximal to the replisome of *FANCJ* null cells. These findings in conjunction with our prior research linking FANCJ to MMR[24] provided evidence that PARP1 and the MMR pathway are dysregulated in FANCJ deficient cells.

Given that MMR and PARP1 bind G4 structures that form in the replisome of FANCJ deficient cells[26–33], we sought to test if the proximity of PARP1 to G4s was also enhanced. First, we confirmed that the BG4 antibody detected greater signal in the RPE1 and U2OS *FANCJ* null cells as well as in WT cells following treatment with the G4 stabilizing agent, pyridostatin (PDS) (Supplementary Fig. 2b, c). Consistent with prior studies[26,27], we find through proximity ligation analysis that PARP1 and G4 DNA are in close proximity (Fig. 2c). Moreover, we observe significantly increased PARP1-G4 PLA signal in *FANCJ* KO relative to WT cells (Fig. 2c). We also observe increased chromatin-bound PARP1 and MSH2 in *FANCJ* null cells or in WT cells treated with PDS suggesting that G4s are bound by PARP1 and MMR proteins (Fig. 2d and Supplementary Fig. 2d). Collectively, these findings demonstrate that FANCJ restricts the proximity of MSH2 with PARP1 and PARP1 with G4s. Thus, we hypothesized that the accumulation of PARP1/MSH2 bound G4s could limit PARP1 activation in *FANCJ* null cells (Fig. 2e).

### FANCJ helicase activity and interaction with MLH1 are required for PARP1 activity during DNA replication

FANCJ helicase activity resolves G4s and FANCJ lysines K141/142 bind G4s[23] as well as mediating an interaction between FANCJ and the MMR protein MLH1[25]. Thus, we expected that mutations disrupting either function would elevate G4s and in turn reduce PARP1 activity. To test this prediction, we generated RPE1 cells with lysine 52 converted to arginine (K52R) to inactivate FANCJ ATPase and helicase activity[34] and lysines 141 and 142 of FANCJ converted to alanine (K141/142A). We observed that the mutant proteins were expressed at similar levels to

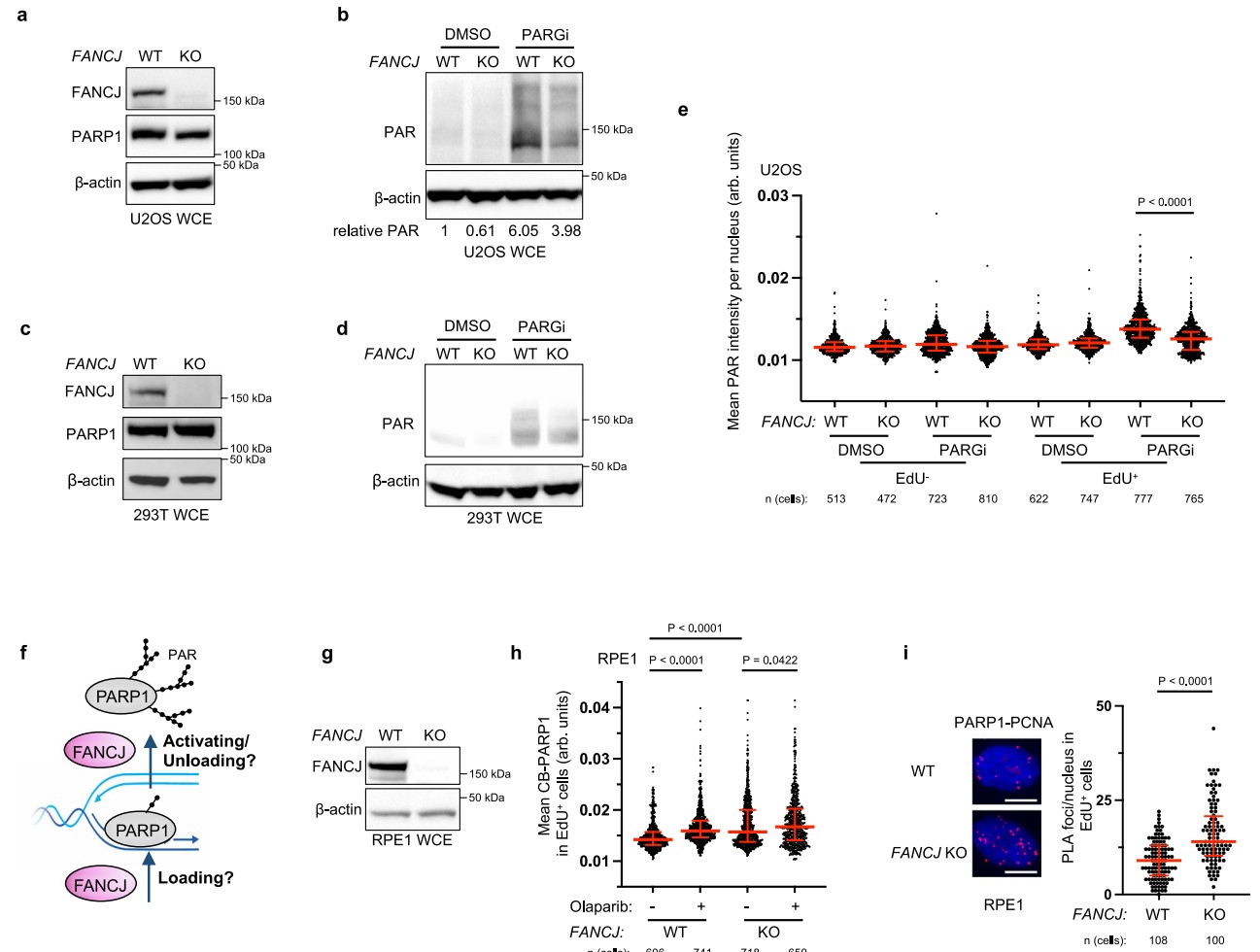

**Fig. 1 | Modest PARPi sensitivity of FANCJ deficient cells is associated with low S-phase PAR and enhanced PARP1 chromatin loading. a** Representative western blot (WB) from two independent experiments for analysis of the expression levels of the indicated proteins from whole cell lysates in untreated WT vs *FANCJ* KO U2OS cells. **b** Representative WB analyzing PAR from whole cell lysates in indicated U2OS cells treated with DMSO or PARG inhibitor (PARGi, 10 μM) for 40 min prior to harvesting to block PAR removal. Mean relative PAR from three independent experiments normalized to β-actin shown. **c** Representative WB from two independent experiments for analysis of the expression levels of the indicated proteins from whole cell lysates in untreated WT vs *FANCJ* KO 293T cells. **d** Representative WB from two independent experiments showing the PAR formation in indicated 293T cells treated with DMSO or PARG inhibitor (PARGi, 10 μM) for 40 min prior to harvesting to block PAR removal. **e** Quantification of mean PAR intensity per nucleus for WT and *FANCJ* KO U2OS cells treated with DMSO or PARGi (10 μM, 30 min) together with EdU incubation. Data are from three independent experiments. Each dot represents one cell. Red bars represent the median ± interquartile range. All statistical analysis according to Kruskal–Wallis test, followed by Dunn's

test. **f** Schematic showing how FANCJ impacts PAR formation during replication: could FANCJ impact PARP1 loading and/or releasing during replication? **g** Representative WB from three independent experiments for analysis of the expression levels of the indicated proteins from whole cell lysates in untreated WT vs *FANCJ* KO RPE1 cells. **h** Quantification of chromatin-bound PARP1 (CB-PARP1) for RPE1 WT and *FANCJ* KO with or without Olaparib treatment (10 μM, 2 h) with EdU incubated in the final 30 min. Data are from three independent experiments. Each dot represents one cell. Red bars represent the median ± interquartile range. All statistical analysis according to Kruskal-Wallis test, followed by Dunn's test. **i** PARP1-PCNA proximity ligation assay (PLA) in untreated RPE1 WT vs *FANCJ* KO cells, with 10 μM EdU incubated for 20 mins. Dot plot shows the number of foci per EdU+ cell and the red bars represent median ± interquartile range from three independent experiments. Scale bars, 10 μm. Statistical analysis according to two-tailed Mann–Whitney test. Representative images shown with PLA foci in red, scale bars 10 μm. For **e**, **h**, and **i**; 5-ethynyl-2'-deoxyuridine (EdU)-positive or EdU+ cells were gated to identify positive EdU incorporation (S-phase). Source data are provided as a Source Data file.

wild-type FANCJ (Fig. 2f). While the mutants enhanced sensitivity to the crosslinking agent, mitomycin C (MMC), PARPi resistance remained similar to the wildtype and null lines (Fig. 2g, h). In addition, we employed a re-expression strategy using either *FANCJ* KO or Fanconi anemia (FA) patient FANCJ deficient (FA-J) cells showing expected MMC sensitivity (Supplementary Fig. 2e–h) and PARPi resistance (Supplementary Fig. 2i). We also confirmed that the KO, knock-in (KI) RPE1 or complemented RPE1 or FA-J cells displayed a significant restart defect following release from aphidicolin (APH) (Supplementary Fig. 2j, k) or hydroxyurea (HU) (Supplementary Fig. 2l, m)[25]. Furthermore, as expected, we observed greater G4 accumulation by BG4 antibody immunofluorescence in the KO and mutant lines as

compared to WT cells (Fig. 2i). Finally, examination of PARP1 activity in unperturbed S phase revealed that cell lines lacking *FANCJ*, its helicase activity, or MLH1 binding had low S-phase PAR compared to *FANCJ* proficient cells (Fig. 2j and Supplementary Fig. 2n). Collectively, these findings suggest that G4s have the potential to restrict PARP1 activity during DNA replication.

## MSH2 promotes G4 formation and limits PARP1 activity during DNA replication

To further test the idea that MSH2 limits G4 processing and in turn PARP1 replication activity, we analyzed the *MSH2* null endometrial adenocarcinoma cell line, HEC59 in which chromosome 2 introduction

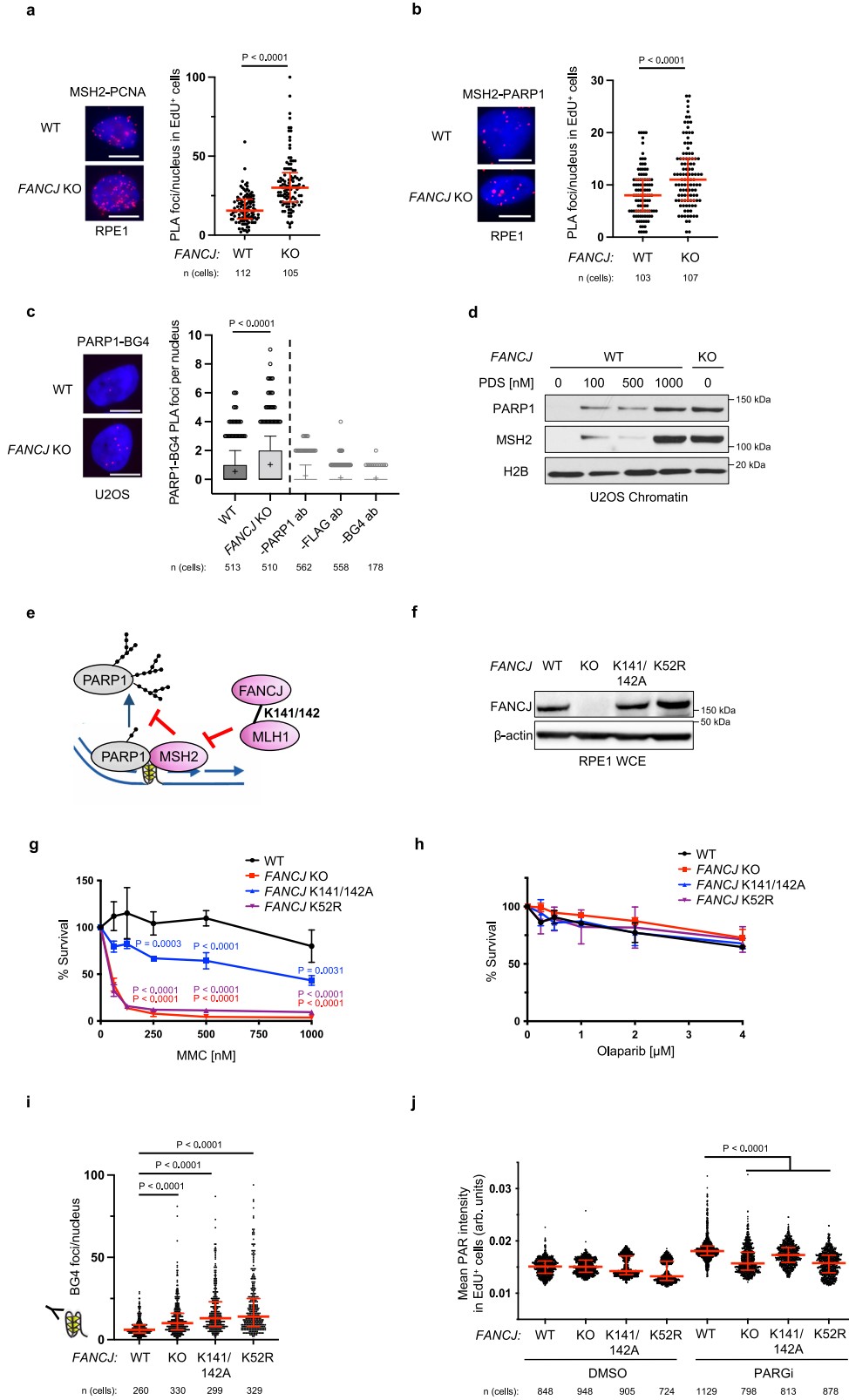

establishes *MSH2* expression[35]. Immunoblot revealed that the HEC59 (MSH2 deficient) and HEC59 + chr2 (*MSH2* proficient) cells display similar levels of total and CB-PARP1 (Fig. 3a). However, *MSH2*-restored HEC59 + chr2 cells had more G4s along with reduced replication proficiency and RPA chromatin loading (Fig. 3b–d and Supplementary Fig 3a, b). Introduction of *MSH2* also reduced PAR levels in both unchallenged conditions and following MMS and $H_2O_2$ treatment,

again with PAR largely restricted to EdU positive cells (Fig. 3e, f and Supplementary Fig. 3c). Consistent with more PARP1 activity in the MSH2 deficient cells, sensitivity to PARPi was greater than the *MSH2* proficient cells whereas as expected the MSH2 deficient cells were more resistant to alkylating agent, N-methyl-N'-nitro-N-nitrosoguanidine (MNNG)[35] (Fig. 3g, h). Similar findings were observed in HeLa cells in which *MSH2* KO maintained elevated RPA but reduced G4s

**Fig. 2 | S-phase PAR requires FANCJ helicase and MLH1 binding activities.**
**a** MSH2-PCNA proximity ligation assay (PLA) in untreated RPE1 WT vs *FANCJ* KO cells with EdU incubated for 20 mins. Dot plot shows the number of foci per EdU+ cell and the red bars represent median ± interquartile range from three independent experiments. Statistical analysis according to two-tailed Mann−Whitney test. Representative images shown with PLA foci in red, scale bars 10 µm. **b** MSH2-PARP1 PLA in untreated RPE1 WT vs *FANCJ* KO cells with EdU incubated for 20 min. Dot plot shows the number of foci per EdU+ cell and the red bars represent median ± interquartile range from three independent experiments. Statistical analysis according to two-tailed Mann-Whitney test. Representative images shown with PLA foci in red, scale bars 10 µm. **c** PARP1-BG4 PLA in untreated U2OS WT vs *FANCJ* KO cells. The mean of the data are represented by a "+", the bounds of box indicate first and third quartile while the whiskers indicate 10th and 90th percentile. Data are from three independent experiments except for the negative BG4 antibody control (1 experiment). Scale bars, 10 µm. Statistical analysis according to two-tailed Mann−Whitney test. Representative images shown with PLA foci in red, scale bars 10 µm. **d** Representative WB from three independent experiments of chromatin bound PARP1 and MSH2 in 24 h PDS treated U2OS WT cells compared to untreated WT and *FANCJ* KO cells. **e** Model showing MSH2 could limit PARP1 activation and be regulated by the FANCJ-MLH1 interaction. **f** WB analysis of FANCJ in the whole cell lysates of FANCJ knock-in mutant cells. Representative from three independent experiments. **g** Cell survival assays for indicated cells under increasing concentrations of mitomycin C (MMC). Dots represent the mean percentage ±SD of survival for each cell line and drug concentration from three independent experiments. Significance was determined by one-way ANOVA followed by Dunnett's test comparing *FANCJ* WT to mutant cells. *P*-value color matches sample in key and compares to WT. **h** Cell survival assays for indicated cells under increasing concentrations of Olaparib. Dots represent the mean percentage ±SD of survival for each cell line and drug concentration from three independent experiments. **i** Quantification of G-quadruplex (BG4 antibody) foci/nucleus in RPE1 mutant cells under untreated growth conditions. Representative from two independent experiments. Statistical analysis according to Kruskal-Wallis test, followed by Dunn's test. **j** Quantification of PAR after 30 min DMSO or PARGi (10 µM) treatment with EdU in EdU+ RPE1 WT, *FANCJ* KO, *FANCJ* K141/142A and K52R cells. Red bars represent the median ± interquartile range. Representative from three independent experiments. Statistical analysis according to Kruskal−Wallis test, followed by Dunn's test. For **a**, **b**, and **j**; 5-ethynyl-2'-deoxyuridine (EdU)-positive or EdU+ cells were gated to identify positive EdU incorporation (S-phase). Source data are provided as a Source Data file.

(Supplementary Fig. 3d−f). PAR levels were also enhanced, along with PARPi sensitivity as compared to *MSH2* proficient controls that had as expected greater MNNG sensitivity (Supplementary Fig. 3g−i). Together, these findings demonstrate that MSH2 aligns with not only enriched G4s but also reduced PARP1 S-phase activity and targetability.

## MSH2 depletion restores PARPi sensitivity and gaps in cells lacking the FANCJ-MLH1 interaction

We next sought to evaluate if G4s and reduced PARP1 activity in cells lacking the FANCJ-MLH1 interaction was caused by MSH2. We first confirmed our previous finding that *MSH2* depletion restored replication restart following release from APH or HU in cells lacking the FANCJ-MLH1 interaction (Fig. 4a and Supplementary Fig. 4a−c)[25]. Moreover, G4s were reduced (Fig. 4b and Supplementary Fig. 4d) and S-phase PAR was elevated in both the RPE1 and FA-J cell systems (Fig. 4c and Supplementary Fig. 4e, f). We also observed that CB-PARP1 was reduced by *MSH2* depletion and accordingly, we observed that PARPi led to a significant re-trapping of PARP1 (Fig. 4d). Finally, we observed that *MSH2* depletion enhanced PARPi-induced replication speed as detected by the lengthening of DNA fibers (Fig. 4e) that can be toxic if discontinuous with gaps[17]. Indeed, we detected replication gaps[17,36] by the presence of RPA loading and S1 nuclease sensitive DNA fibers (Fig. 4e and Supplementary Fig. 4g). Thus, in cells without the FANCJ-MLH1 interaction, MSH2 limits the capability of PARPi to promote speed and gaps indicating that PARP1 activation may be deficient. Indeed, *MSH2* depletion enhanced PARPi sensitivity (Fig. 4f and Supplementary Fig. 4h, i), consistent with restored PARP1 catalytic activity. Notably, *FANCJ* null cells have a more muted response to *MSH2* depletion (Supplementary Fig. 4i) as found with DNA interstrand crosslinking agents[25], suggesting that this enhanced sensitivity requires FANCJ. Thus, it appears that *MSH2* depletion restores FANCJ function in the FANCJ-MLH1 mutant cells consistent with MSH2 and FANCJ having opposing activities that are regulated by the FANCJ-MLH1 interaction.

## G4 "sequestered" and canonical "trapped" PARP1 are distinct and provide insight into PARPi toxicity in BRCA deficient cells

*FANCJ* loss limits PARP1 activity which along with PARP1 trapping is thought to underlie the toxicity of PARPi. Thus, we sought to understand if PARP1 bound to G4s[28]. "G4-sequestered PARP1" was similar to canonical trapped PARP1 induced by PARPi (Fig. 5a). Canonical PARP1 trapping is detrimental in the context of *XRCC1* null cells[37] as validated by the finding that *XRCC1* null cells are hypersensitive to PARPi yet insensitive to loss of PARP1 catalytic activity by *PARP1* deletion

(Fig. 5b, c and Supplementary Fig. 5a, b), as reported[38,39]. If *FANCJ* depletion enhanced canonical trapped PARP1, *FANCJ* depletion could also sensitize *XRCC1* null cells. However, *FANCJ* depletion did not reduce the colony formation of the *XRCC1* null cells and instead modestly elevated PARPi resistance (Fig. 5b, c). Moreover, FANCJ-deficient cells did not gain fitness following *PARP1* deletion (Fig. 5d) suggesting that G4-sequestered PARP1 was not inherently toxic. However, as expected, *BRCA1* depletion in *PARP1* KO cells reduced clonogenic efficiency akin to the sensitivity detected with PARPi treatment (Fig. 5d) consistent with the dependency of BRCA1 deficient cells on PARP1[2,40–43]. Functional depletion of *FANCJ* or *BRCA1* was confirmed by immunoblot and by the sensitivity to PARPi versus cisplatin, respectively (Supplementary Fig. 5c, d). Overall, these findings indicated that *FANCJ* depletion creates a form of chromatin-bound PARP1 that is not inherently toxic at least in FANCJ deficient or *XRCC1* null cells.

Given that PARP1 binds G4s[28] and is distinct from trapped PARP1, we sought to determine if this insight could be leveraged to clarify how PARPi is toxic in BRCA1 deficient cells. Indeed, PARP1 sequestering that limits PARP1 replication activity could be toxic in settings in which this PARP1 activity is essential. A cell system that could be dependent on PARP1 S-phase activity is BRCA1 deficient cells that display elevated PARP1 S-phase activity[17]. Moreover, BRCA1 deficient cells display increased PARylation when subjected to MMS and/or MMS followed by $H_2O_2$ as compared to wildtype or *XRCC1* null cells that exhaust this activity[37] (Supplementary Fig. 5e, f) indicating that PARP1 is readily activated in BRCA1 deficient cells. *BRCA1* and *FANCJ* DKO cells were viable; however, colony plating revealed that fitness was significantly curtailed compared to either single deficiency (Fig. 5e and Supplementary Fig. 5g) as further confirmed by siRNA depletion of *BRCA1* or *FANCJ* in the single KO lines compared to WT (Supplementary Fig. 5h). This finding was similar to the greater than expected loss of viability between *FANCJ* and *BRCA1* dual deletion as compared to each single deletion[18]. Collectively, these findings suggest that loss of PARP1 replication activity due to PARPi, *PARP1* deletion or *FANCJ* loss is toxic to BRCA1 deficient cells (Fig. 5f) predicting that PARPi efficacy in *BRCA* mutant cancer derives from loss of PARP1 S-phase activity and not PARP1 trapping.

## Discussion

BRCA1 and FANCJ are known to be involved in DNA double-strand break repair by homologous recombination, fork protection, and Fanconi anemia pathways. Moreover, mutations in these genes increase the risk of developing hereditary breast and ovarian

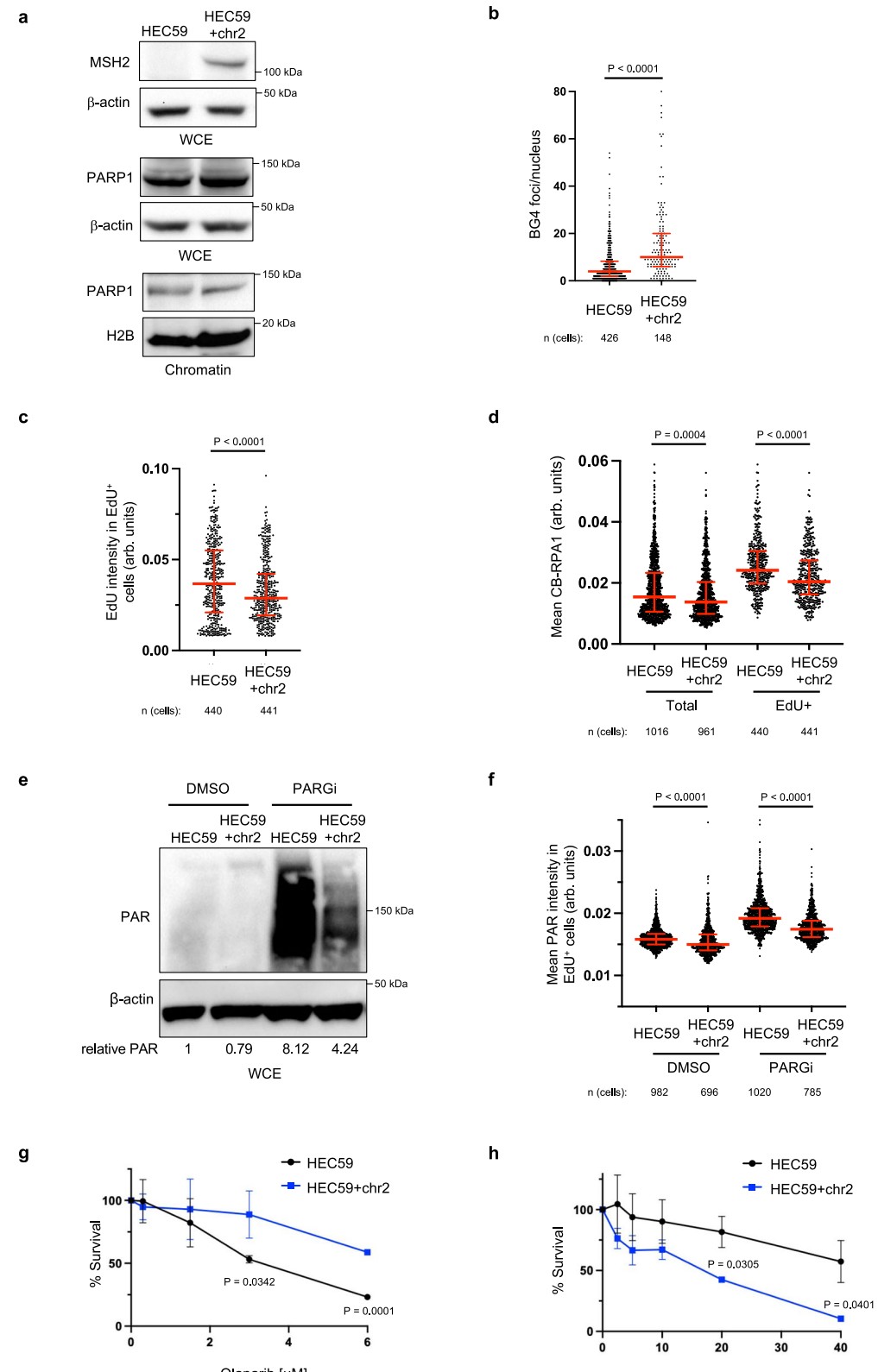

cancer[14–16,44]. Based on these common functions, it was expected that akin to BRCA1 deficient cells, FANCJ deficient cells would be robustly sensitive to PARP inhibitors (PARPi), but they are not[17]. The present study aimed to explore the distinct role of FANCJ in PARPi-induced gaps and sensitivity and to gain insight towards the relationship between gaps and PARP1 function and/or trapping. We find that without FANCJ, PARP1 S-phase activity is abnormally low because

PARP1 is "sequestered" in chromatin which is distinct from canonical chromatin trapping of PARP1. Accordingly, *FANCJ* depletion can elevate PARPi resistance in cells with sensitivity to canonical trapped PARP1. However, similar to PARPi or *PARP1* deletion, *FANCJ* deletion in BRCA1 deficient cells is toxic. These findings highlight that the common feature of toxicity in BRCA1 deficient cells is loss of PARP1 replication activity as opposed to PARP1 trapping. Given that BRCA1

**Fig. 3 | MSH2 interferes with PARP1 activation. a** Representative WB from two independent experiments from whole cell lysates and chromatin fractionations showing indicated proteins in HEC59 vs HEC59 + chr2 cells under untreated growth conditions. **b** Quantification of G-quadruplexes (BG4 antibody) in untreated HEC59 vs HEC59 + chr2 cells from three independent experiments. Red bars represent the median ± interquartile range, each dot represents one cell. Statistical analysis according to two-tailed Mann-Whitney test. **c** Quantification of mean EdU intensity from EdU⁺ cells labeled for 40 min from three independent experiments. Red bars represent the median ± interquartile range, each dot represents one cell. **d** Quantification of chromatin bound RPA1 (CB-RPA1) for the indicated cells with EdU incubated for 40 min from three independent experiments. Red bars represent the median ± interquartile range, each dot represents one cell. Statistical analysis according to Kruskal-Wallis test, followed by Dunn's test. **e** Representative WB for the PAR formation in indicated cells treated with DMSO or PARGi (10 μM) for 40 min prior to harvesting. Mean relative PAR from three independent experiments normalized to β-actin shown. **f** Quantification of PAR after 30 min DMSO or PARGi (10 μM) treatment in the indicated EdU⁺ cells. Each dot represents one cell from three independent experiments. Red bars represent the median ± interquartile range, each dot represents one cell. All statistical analysis according to Kruskal-Wallis test, followed by Dunn's test. **g, h** Cell survival assays for the indicated cells under increasing concentrations of Olaparib and *N*-methyl-*N*′-nitro-*N*-nitrosoguanidine (MNNG). Dots represent the mean percentage ±SD of survival for each cell line and drug concentration from three independent experiments. Significance was determined by unpaired *t*-test (two-tailed, unequal variance). For **c**, **d**, and **f**; 5-ethynyl-2′-deoxyuridine (EdU)-positive or EdU⁺ cells were gated to identify positive EdU incorporation (S-phase). Source data are provided as a Source Data file.

deficient cells have elevated PARP1 replication activity, our findings support that loss of PARP1 S-phase activity is lethal in BRCA1 deficient cells because they rely on this activity to mitigate replication problems.

This detailed elucidation of the role of FANCJ in PARP1 activity, MSH2 localization, and G4 stabilization provides insight into cellular replication processes that impact PARPi efficacy and gap induction (Supplementary Fig. 5i). FANCJ, although not required for PARP1 chromatin localization, is critical in maintaining proper PARP1 activity levels and MMR pathway regulation. FANCJ deficient cells, observed in various types including RPE1, 293T and U2OS, exhibited conspicuously low PARP1 S-phase activity. FANCJ deficiency did not inherently prevent PARP1 activation in response to DNA damage, indicated by maintained PARP1 activity in cells treated with DNA damaging agents like MMS and $H_2O_2$. Instead, our findings reveal that a FANCJ-MLH1 interaction counters MSH2-bound G4s that sequester and restrict PARP1 S-phase activation. Consistent with this model, FANCJ deficient cells have elevated G4s, that are substrates for PARP1 and MSH2[26–33], and correspondingly greater proximity between G4s and PARP1. Moreover, MSH2 deficient cells display more G4s and PARP1 activity compared to MSH2 proficient cells, which showed reduced PARP1 S-phase activity and targetability. Our previous studies have proposed that FANCJ restricts the MSH2-MSH6 complex since both MSH2 and MSH6 are elevated in the replisome of *FANCJ* null cells, and depletion of *MSH2* or *MSH6*, but not *MSH3*, rescues defects associated with FANCJ deficiency[16,25]. Moreover, given that *MSH2* depletion restores PARP1 activity in cells lacking the FANCJ-MLH1 interaction, but not in cells without *FANCJ*, G4 unfolding ultimately requires FANCJ and may be opposed by MSH2 unless coordinated with MLH1. The MMR pathway is present at the replisome[45], however in *FANCJ* null cells, MSH2/MSH6 becomes further enriched[16], thus we propose that this complex is uniquely restricted by FANCJ.

While our understanding of how PARP1 is activated during an undisturbed S phase to participate in backup lagging strand synthesis remains limited, our findings suggest that G4 structures and MMR possess the potential to inhibit this activity, thereby conferring resistance to PARPi. Foreseeably, the PARP1 auto-inhibitory domain is not displaced when PARP1 binds negatively charged G4s as it would be at a DNA break. Indeed, FANCJ unfolds proteins, and unfolding of the PARP1 helical region is associated with its catalytic activation[46–48], leading to the possibility that FANCJ unfolds and activates PARP1. FANCJ may also create a G4 structure that is amenable to PARP1 activation. Prior to unfolding G4s, FANCJ stabilizes G4s, and G4 stabilization is associated with ssDNA gaps and PARP1 activation[23,49,50]. Moreover, while like MMR, PARP1 has affinity for G4s, only specific G4s activate PARP1[29–32]. Furthermore, FANCJ displacement of MSH2 appears fundamental for PARP1 replication function. Intriguingly, MMR impacts centromeric DNA replication by binding secondary structures that limit RPA and checkpoint induction[51] consistent with MMR occluding ssDNA. It remains to be determined whether a MMR-FANCJ pathway engagement and S-phase PAR levels

can ultimately serve as a pharmacodynamic marker of PARPi response[52]. However, the dependence of BRCA1 deficient cancer cells on PARP1 S-phase activity predicts that loss of this activity such as by targeting FANCJ will be a therapeutic option.

The finding that *FANCJ* null cells readily tolerate the enriched PARP1 in chromatin also provides insight that not all "trapping" is the same. Indeed, canonical trapping underlies the sensitivity of XRCC1 deficient cells to MMS[37]. Consistent with this interpretation, we and others find these cells gain fitness to MMS treatment as well as resistance to PARPi upon *PARP1* loss[37]. Given that XRCC1 deficient cells also gain PARPi resistance with *FANCJ* loss, we can surmise that canonical trapping is also reduced in this setting. Indeed, PARPi sensitivity in several genetic contexts is reversed by *PARP1* deletion consistent with a trapping model[18,53–55]. Likewise, in the context of *BRCA1* mutant cells in which residual BRCA function tolerates *PARP1* loss, PARPi toxicity can be suppressed by mutations that reduce PARP1 trapping[56–58]. Moreover, cancer cells with distinct *BRCA* mutations, gain resistance by loss of PARP1[40,58].

By contrast, when *PARP1* loss causes a major reduction in fitness similar to PARPi, it is likely toxic due to the disruption of PARP1 S-phase activity; therefore, PARP1 trapping may be incidental to PARP1 catalytic inhibition. Indeed, the finding that *FANCJ* loss is toxic with *BRCA1* loss highlights that loss of PARP1 activity could be the fundamental issue. This possibility is supported by the finding that BRCA-deficient cells are reported to be sensitive to either PARPi or *PARP1* loss[2]. Our own findings also align with this conclusion. The idea that PARP1 activity in S phase maintains the homeostasis in BRCA deficient cells is furthered by the finding that PARP1 activity is elevated as shown previously and herein[17]. Furthermore, the role of BRCA proteins in preventing replication-associated gaps sheds new light on why PARP1 activity could be essential. Dual loss of *BRCA* and *PARP1*, and the combined replication dysfunction, creates a toxic level of gaps that grossly limits cell fitness[13,17,59]. This phenomenon has also been observed with combined loss of *POLQ* (Polθ) and *BRCA*[60–62]. The theory postulating that the toxicity of PARPi in BRCA deficient cells is attributed to the capacity of the drug to trap PARP1, substantiated by discoveries indicating that more potent PARP trappers exhibit increased toxicity to these cells, has recently faced skepticism, spurred by emerging evidence suggesting that more potent trappers inherently possess enhanced catalytic inhibitory strength[10,63,64].

In summary, our research sheds light on the complex mechanisms underlying PARP inhibitor sensitivity in different genetic contexts and highlights that in BRCA deficient cells, the anti-cancer mechanism is due to loss of PARP1 S-phase activity. We believe that these findings have significant implications for the development of targeted therapies and precision medicine approaches for treating cancer. Furthermore, our findings highlight that depending on the underlying mechanism of action, PARPi resistance will also evolve in distinct ways.

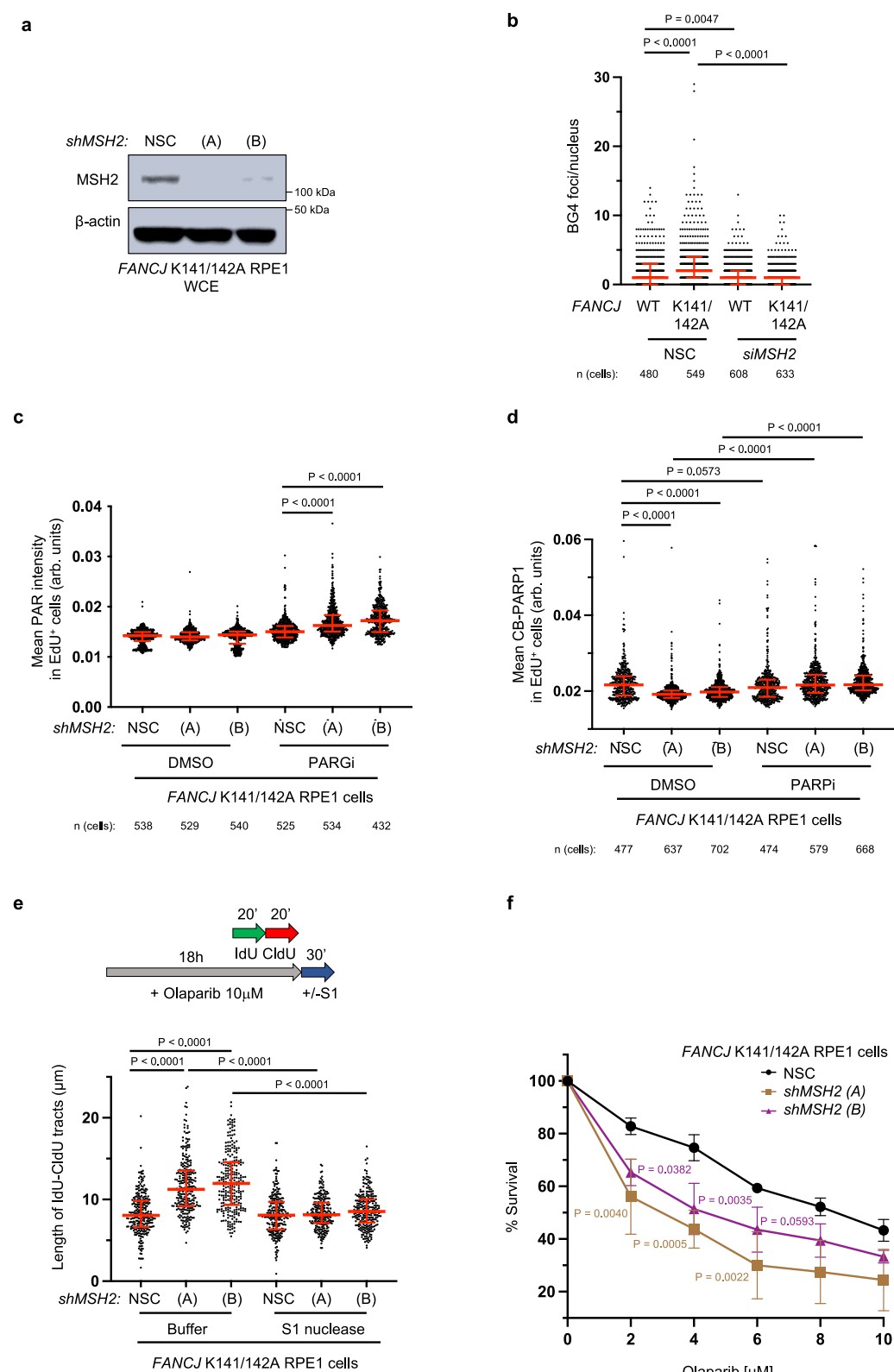

## Methods

### Cell lines and related reagents

Human RPE1-hTERT *TP53*[-/-] (WT, parental) and *BRCA1* KO cells are from Dr. Daniel Durocher. The HeLa WT and *MSH2* KO cells, in addition to the human endometrial HEC59 and HEC59 + chr2 cell lines, were originally from the Dr. Christopher Heinen lab. The RPE1, 293T, U2OS and HeLa-derived cell lines were grown in DMEM supplemented with 10%

fetal bovine serum (FBS, Sigma-Aldrich), 1% penicillin/streptomycin (Gibco) and 1% sodium pyruvate (only for RPE1). The generation of the *FANCJ* KO RPE1-hTERT (*TP53*[-/-]), 293T and U2OS cell lines were described before[16,17]. The *BRCA1* and *FANCJ* double KO cells were generated similarly as before. *PARP1* and *XRCC1* related WT, KO, and DKO RPE1 cells were from the Dr. Keith Caldecott lab[37] and grown in DMEM/F12 (Gibco), 10% fetal bovine serum (FBS, Sigma-Aldrich), 1% penicillin/

**Fig. 4 | The FANCJ-MLH1 interaction promotes PARP1 activation by restricting MSH2. a** Representative WB from three independent experiments for the indicated proteins from whole cell lysates from RPE1 cells expressing small hairpin RNA (shRNA) against non-silencing control (NSC) and two shRNAs targeting *MSH2* (A) and (B). **b** Quantification of G-quadruplexes (BG4 antibody) in the indicated RPE1 cells with siRNA under untreated growth conditions. Each dot represents one cell from three independent experiments. Red bars represent the median ± interquartile range. Statistical analysis according to Kruskal-Wallis test, followed by Dunn's test. **c** Quantification of PAR after 30 min DMSO or PARGi (10 μM) treatment in the indicated EdU⁺ RPE1 cells. Each dot represents one cell from three independent experiments. Red bars represent the median ± interquartile range. Statistical analysis according to Kruskal–Wallis test, followed by Dunn's test. **d** Quantification of CB-PARP1 in the indicated cells with or without Olaparib treatment (10 μM, 6 h), with EdU incubated at the final 40 min. Each dot represents

one cell from three independent experiments. Red bars represent the median ± interquartile range. Statistical analysis according to Kruskal–Wallis test, followed by Dunn's test. **e** Schematic and quantification of the S1 nuclease DNA fiber assays for the length of dual-color tracts in indicated cells following Olaparib treatment (10 μM, 18 h). Each dot represents 1 fiber; data are from three independent experiments. Red bars represent the median ± interquartile range. Statistical analysis according to Kruskal–Wallis test, followed by Dunn's test. **f** Cell survival assays for indicated RPE1 cells under increasing concentrations of Olaparib. Dots represent the mean percentage ±SD of survival for each cell line and drug concentration from four independent experiments. Significance was determined by one-way ANOVA followed by Dunnett's test, *P*-value color matches sample in key and compares to NSC. For **c** and **d**; 5-ethynyl-2′-deoxyuridine (EdU)-positive or EdU⁺ cells were gated to identify positive EdU incorporation (S-phase). Source data are provided as a Source Data file.

streptomycin (Gibco). FA-J (EUFA30-F) cells are from Dr. Hans Joenje. The HEC59 lines and FA-J cells grown were cultured in DMEM supplemented with 15% FBS. For the RPE1-hTERT (*TP53⁻/⁻*) *FANCJ* gene knock-in cell lines, the desired *FANCJ* gene variants (K141/142A, K52R) were introduced in the RPE1-hTERT (*TP53⁻/⁻*) cells according to the RNP CRISPR approach of IDT. Detailed steps, sequences of PCR primers, sgRNA, and ssODN repair templates can be found in Supplementary table 1. The expression of FANCJ variants were confirmed by immunoblot analysis.

## Drugs and other reagents
The following drugs were used in the course of this study: PARP inhibitor Olaparib (SelleckChem AZD-2281 S1060), cisplatin (Sigma-Aldrich P4394), camptothecin (CPT, Sigma-Aldrich C9911), methyl methanesulfonate (MMS, Sigma-Aldrich 129925), N-Methyl-N′-nitro-N-nitrosoguanidine (MNNG, GLPBIO GC36598-1), PARG inhibitor (PDD 0017273, Tocris 5952), aphidicolin (Sigma A0781), hydroxyurea (Sigma-Aldrich H8627), hydrogen peroxide solution (H₂O₂, H1009, Sigma-Aldrich), mitomycin C (MMC, MA4287, Sigma-Aldrich), dimethyl sulfoxide (DMSO, Sigma-Aldrich D5879). All drugs were directly used or prepared per the manufacturer's instructions. Other reagents used in this study included 5-chloro-2′-deoxyuridine (CldU, Sigma-Aldrich C6891), 5-Iodo-2′-deoxyuridine (IdU, Sigma-Aldrich I7125). Concentration and duration of treatment are indicated in the corresponding figures and sections.

## Immunoblotting
Cells were harvested, lysed, and processed for western blot analysis as described previously using 150 mM NETN lysis buffer [20 mM Tris; (pH 8.0), 150 mM NaCl, 1 mM EDTA, 0.5% NP-40, and Halt Protease inhibitor cocktail (Thermo Fisher Scientific 78440)][17]. For PAR blotting, the addition of 10 μM PARGi (Tocris) and 10 μM Olaparib (Selleckchem) was added to the lysis buffer. For cell fractionation, cytoplasmic and soluble nuclear fractions were isolated with the NE-PER Kit (Thermo Fisher Scientific 78835) according to the manufacturer's protocol; to isolate the chromatin fraction, the insoluble pellet was resuspended in RIPA buffer (Cold Spring Harbor Protocol) and followed by 15 min sonication by Diagenode bioruptor with medium power for 30 s on and 30 s off at 4 °C. Proteins were separated using SDS−PAGE and electro-transferred to nitrocellulose or polyvinylidene difluoride membranes. Membranes were blocked in 5% non-fat dry milk (NFDM) in Tris-buffered saline with 0.1% Tween-20 and incubated with primary antibody overnight at 4 °C. Primary antibodies for western blot analysis included anti-FANCJ 1:500 (E67 from Cantor lab), anti-β-actin 1:5000 (Sigma-Aldrich A1978), anti-Tubulin 1:2000 (Abcam ab6160), anti-PAR 1:10,000 (poly-ADP-ribose binding reagent, Millipore Sigma MABE1031), anti-PARP1 1:1000 (Abcam ab227244), anti-H2B 1:1000 (Cell Signaling Technology 8135), anti-MSH2 1:2000 (Abcam ab52266), anti-XRCC1 1:1000 (Abcam ab134056). Secondary antibodies include

ECL anti-rabbit IgG, HRP-linked whole antibody 1:10,000 (from donkey, GE Healthcare NA934) and ECL anti-mouse IgG, HRP-linked F(ab′)₂ fragment 1:10,000 (from sheep, Thermo Fisher Scientific NA9310) All antibodies were used within the range of suggested dilution. Membranes were washed, incubated with corresponding horseradish peroxidase-linked secondary antibodies (Amersham, GE Healthcare) for 1 h at room temperature (RT) and detected by chemiluminescence imaging system (Bio-Rad) or Kodak X-OMAT 2000A Film Processor. Quantification of immunoblots performed with ImageJ (Fiji v2.1.0) by normalizing PAR to β-actin and calculating relative PAR levels with respect to the WT untreated sample.

## Plasmids and RNA interference
Lentiviral production was described previously in detail[65]. Similarly here, the mutant clones of *FANCJ* K141/142A and K52R were individually generated using site-directed mutagenesis (Genscript) with PMT-BRD025 *FANCJ* WT (wild-type) as the template, and were sequence verified (Genscript Piscataway). Details of standard virus production pipelines can be found at the Broad Institute's Genetic Perturbation Platform website (https://portals.broadinstitute.org/gpp/public/). Viruses for the mutant and WT *FANCJ* were produced in 96-well plates using HEK293T cells transfected with packaging vector psPAX2 (100 ng), envelope plasmid CMV-VSVG (10 ng), and respective PMT-BRD025 *FANCJ* mutant plasmid (100 ng). Lentiviral-containing supernatants were harvested 31 hrs post-transfection and stored in polypropylene plates at −80 °C until use.

Stably transduced cells were generated by infection with pLKO.1 vectors containing shRNAs against non-silencing control (NSC) or one of the shRNAs against corresponding genes: *MSH2* includes (A) 5′-AGCAAGCTCTGCAACATGAAT-3′, (B) 5′-TTACCTTCATTCCATTACTGG-3′. The information was obtained from Dharmacon website (https://horizondiscovery.com), and the shRNAs were obtained from the University of Massachusetts Chan Medical School shRNA core facility. Cells were selected by puromycin for 3–5 days before experiments were carried out. Transfection of siRNA in RPE1 cells was performed with Lipofectamine 3000 (Thermofisher) according to the manufacturer's instructions. Briefly, cells were plated in 8-well slides and each well was transfected with 1 μl of Lipofectamine 3000 and contained a final siRNA concentration of 36 nM in a total volume of 0.5 mL. Assays were performed at 48 h post transfection. siRNA used were ON-TARGETplus Human *MSH2* (4436) SMARTpool (L-003909) and ON-TARGETplus non-targeting pool (D-001810). For studies involving depletion of *BRCA1* or *FANCJ*, cells were plated in 6-well dishes. The next day, each well received 3 μL of DharmaFECT 1 Transfection Reagent (Horizon Discovery) and a final siRNA concentration of 25 nM according to the manufacturer's instructions. Dharmacon siRNA used were *siLuciferase* (D-002050-01-20 or D-001210-02-50), *siBRCA1* SMARTpool (M-003461-00-05), and *siFANCJ* (5′ GUACAGUACCCCA CCUUAU 3′).

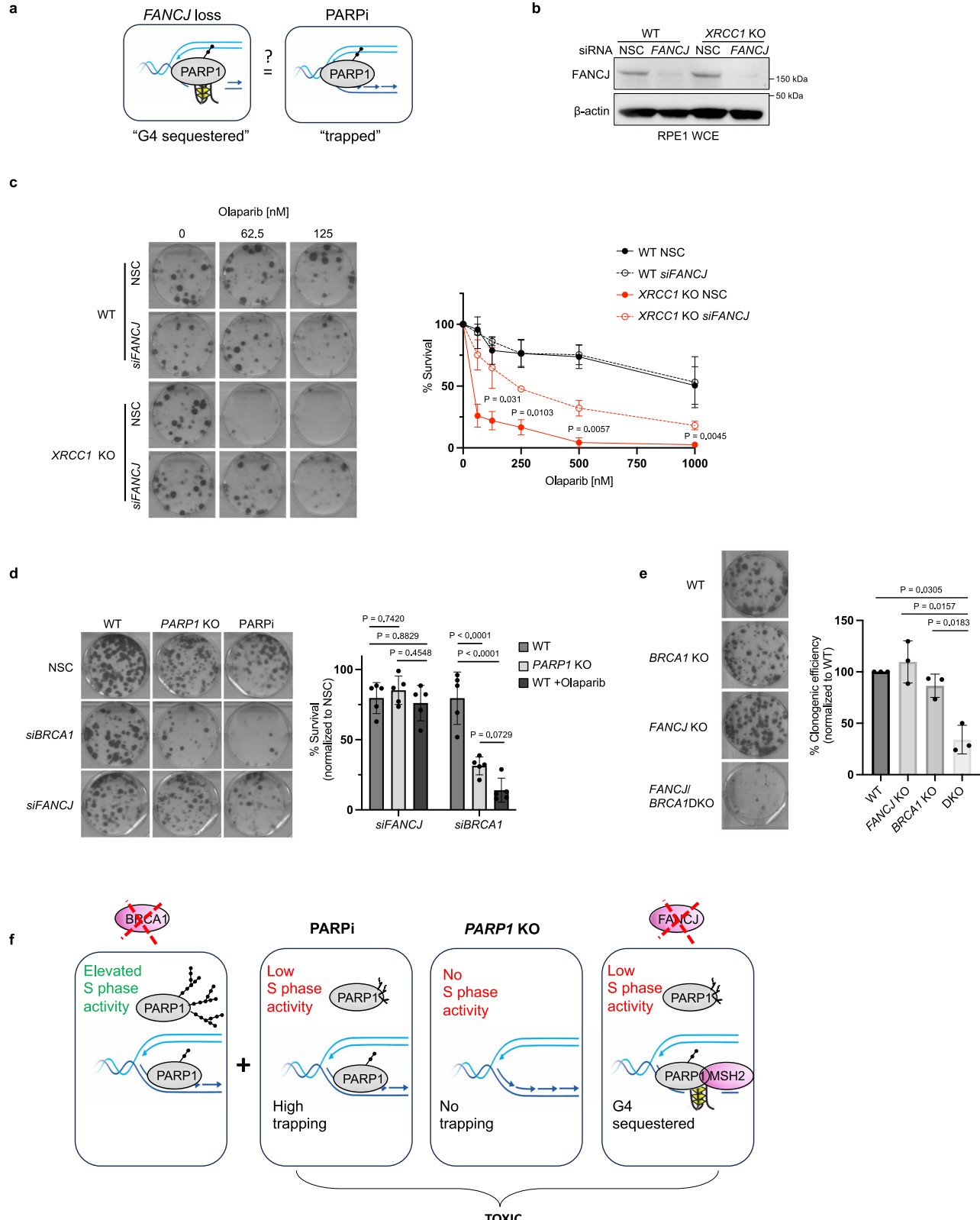

## Viability assays

Cells were seeded onto 96-well plates (300 cells per well, performed in biological triplicates for each experiment group) and incubated overnight. The next day, cells were treated with increasing doses of drugs as indicated in corresponding figures and maintained in complete media for 5– days. For survival assays with MMC and MNNG, the chemicals were incubated with cells the day after seeding for 1 h in serum free medium before washing 2x with PBS and replacing with complete medium for outgrowth. Percentage survival was measured photometrically using CellTiter-Glo 2.0 (Promega G9242) viability assay in a microplate reader (Beckman Coulter DTX 880 Multimode Detector). For clonogenic survival assays, 250–500 cells per well were seeded into 6-well plates. The next day, the media was replaced with media containing treatments or vehicle and incubated for 10 to 14 days.

**Fig. 5 | G4 "sequestered" PARP1 is distinct from "trapped" PARP1 and provides insight that loss of S-phase activity underlies PARPi toxicity in BRCA1 deficient cells. a** Model depicting the question: Is PARP1 trapping distinct from G4 sequestering? **b** Representative WB from two independent experiments to analyze of the expression levels of the indicated proteins from whole cell lysates for the indicated cells lines and knock down reagents. **c** Representative images and quantification of clonogenic survival assays of the indicated RPE1 cell lines and siRNA treatments with increasing doses of Olaparib from three independent experiments. Dots represents the mean percentage ± SD for a given cell line, knockdown reagent and Olaparib concentration. Significance determined by unpaired t-test (two-tailed, unequal variance) comparing *XRCC1* KO NSC to *siFANCJ*. **d** Representative images and quantification of clonogenic assays for RPE1 WT and *PARP1* KO cells with indicated siRNA, KO cell and Olaparib treatment (1 μM). Mean

survival percentages normalized to NSC from 5 independent experiments. Dots represent each individual experiment while the top of the bar indicates the mean percentage ± SD. Significance determined by two-way ANOVA followed by Tukey's test. **e** Representative images and quantification of clonogenic assays for the indicated cells. DKO double knockout. Mean percent clonogenic efficiencies (normalized to WT) from three independent experiments. Dots represent each individual experiment while the top of the bar indicates the mean percentage ± SD. Significance determined by one-way ANOVA (unequal variance) followed by Dunnett's test. **f** Model of proposed findings: PARPi toxicity in BRCA1 deficient cells derives from an inability of PARP1 to function in S-phase resulting in the accumulation of replication-associated ssDNA lesions and PARP1 trapping. Loss of *FANCJ* in BRCA1 deficient cells sequesters PARP1 on G4s prohibiting PARP1 from efficiently functioning during replication. Source data are provided as a Source Data file.

Percentage survival was determined by manual colony counting of a size at least 50 cells or more[66] after staining with crystal violet staining solution (0.05% w/v crystal violet, 1% formaldehyde, 1% MeOH, in 1x PBS) (Sigma-Aldrich C0775).

## Immunofluorescence

For poly(ADP-ribose) or PAR, cells cultured on coverslips were fixed with 4% formaldehyde in PBS for 10 min at RT and subsequently permeabilized by a 5 min incubation in ice-cold methanol/acetone solution (1:1). After blocking the cells with 10% fetal calf serum for 30 min (alternatively add 3% BSA), coverslips were incubated with the primary antibody anti-PAR polyclonal antibody (1:500 Trevigen 4336-BPC-100) at 37 °C for 1 h. Followed by PBS washing, cells were then incubated with the appropriate fluorescently labeled secondary antibody for 1 h at RT. To label EdU, a click reaction (100 mM Tris pH 8, 100 mM CuSO4, 2 mg/ml sodium-L-ascorbate, 10 mM Alexafluor 488 azide) was performed for 30 min. Coverslips were then washed, stained with DAPI (1 mg/ml in PBS, Thermo Fisher Scientific D1306) for 30 min and mounted using VECTASHIELD mounting media (Vector Laboratories H-1200).

For chromatin bound proteins, cells on coverslips were plated on ice for 0.5–1 min before pre-extracted by ice-cold PBS + 0.5% Triton X-100 for 5 min. Then, cells were fixed by 3%paraformaldehyde/2% sucrose for 10 min at RT. Cells were washed twice with PBS-T (0.01% Tween) and incubated with primary antibodies (anti-PARP1 antibody 1:500, Abcam ab227244; anti-MSH2 antibody 1:200, Abcam ab52266; anti-RPA1 antibody 1:500, Cell Signaling Technology 2267) in DMEM + 10% FBS (alternatively add 3% BSA) at 37 °C for 1 h. After 3x PBS-T washing, coverslips were incubated with appropriate secondary antibodies (Alexa Fluor 488 goat anti-Mouse A-11001 and Alexa Fluor 568 goat anti-Rabbit A-11011) in DMEM + 10% FBS and DAPI. EdU labeling was performed as described above. Finally, after washing with PBS-T (x3), coverslips were mounted with Prolong (Invitrogen P36930). For all assays above, images were collected by fluorescence microscopy (Axioplan 2 imaging and Axio Observer, Zeiss) at a constant exposure time in each experiment. Representative images were processed by ImageJ software. Mean intensity of immunofluorescence for each nucleus were measured with Cell Profiler software version 3.1.5 from Broad Institute.

For BG4 Immunofluorescence, RPE1 and HEC59 cells were cultured overnight in 8-well slides (Nunc Lab-Tek II Chamber Slide) with 50,000 cells per well. Cells were washed in PBS and then fixed in 4% paraformaldehyde for 10 min at room temperature. Following three PBS washes, cells were permeabilized with 0.1% Triton X-100 in PBS. Cells were treated with 0.24 mg/ml Monarch RNase A (New England Biolabs) for 1 h at 37 °C. After a PBS wash, cells were blocked with 0.5% goat serum, or alternatively 0.5% FBS, in PBS for 1 h at 37 °C, followed by incubation with BG4 antibody at a 1:100 dilution in 0.1% Tween20 in PBS (PBST) and 0.5% goat serum at 37 °C for 1 h. The BG4 antibody was prepared according to ref. 67, DYKDDDDK Anti-FLAG antibody (cat. no. 14793 S, Cell Signaling Technology) was used at 1:800 in 0.1% PBST

and 0.5% goat serum for 1 h at 37 °C. Following three washes in 0.1% PBST, cells were incubated with 1:1000 Alexa Fluor 594 goat anti-rabbit secondary antibody (cat. no. A-11012, Life Technologies) in 0.1% PBST and 0.5% goat serum for 1 h at room temperature. Finally, cells were overlayed with VECTASHIELD Antifade Mounting Media with DAPI (cat. no. H-1200-10, Vector Laboratories) before sealing. Fluorescent foci were imaged using a Zeiss LSM 710 confocal microscope (objective magnification: ×40, oil). Each experiment was performed with three biological replicates. N = 5 images were analyzed for each condition using a custom ImageJ 2.9.0 script and GraphPad Prism 9.4.1. Where mean fluorescence intensity is reported, images were taken at ×63 with fluorescence microscopy (Axioplan 2 imaging and Axio Observer, Zeiss) and analyzed with Cell Profiler software version 4.2.1 from the Broad Institute.

## Proximity ligation assay

Cells were seeded on 18 mm × 18 mm coverslips and the following day they were pulsed with 10 mM EdU for 20 min. After the EdU pulse, cells were initially pre-fixated with 0.1% formaldehyde in PBS for 2 min at RT. Following three PBS washes, the samples were pre-extracted with CSK buffer on ice for 5 min and then washed with PBS before fixing with 3.7% formaldehyde at RT for 15 min. Coverslips were then washed with PBS and stored O/N at 4 °C. The following day cells were post-fixated and permeabilized with methanol for 20 min at −20C. The samples were washed 3X with PBS and blocked for 30 min with 3% BSA in PBS. To label EdU, a click reaction (100 mM Tris pH 8, 100 mM CuSO4, 2 mg/ml sodium-L-ascorbate, 10 mM biotin-azide) with Alexafluor 488 azide was performed for 30 min. Slides were then incubated with primary antibodies for 1 h at 37 °C (1:250 mouse anti-MSH2 Abcam ab52266; 1:500 rabbit anti-PARP1 Abcam ab227244; 1:500 rabbit anti-PCNA Abcam ab18197, 1:500 mouse anti-PCNA Abcam ab29 diluted in blocking solution). After antibody incubation, coverslips were washed 2X with Buffer A for 5 min at RT (Duolink kit DUO92101). Each coverslip was then incubated for 1 h at 37 °C with Duolink PLA probes (Thermo Fisher Scientific) diluted in blocking solution. After 2X washes with Buffer A for 5 min at RT, probes were ligated for 30 min at 37 °C and amplified by polymerase reaction for 100 min at 37 °C. Coverslips were then washed 2X with Buffer B for 5 min at RT (Duolink kit) and then mounted with DAPI on microscope slides. Images were acquired by fluorescence microscopy (Axioplan 2 imaging and Axio Observer, Zeiss) with a 63X objective. Deconvolution of the images was done using the ImageJ software. The number of foci in each cell was counted with Cell Profiler software and the statistical analysis was performed using GraphPad Prism (v9.2.0).

For the BG4-PARP1 PLA, the above protocol was modified to include the following steps. After post-fixation and permeabilization with methanol for 20 min at −20C, the samples were washed three times for 5 min each with PBS-0.01%Tween and incubated with 0.24 mg/ml Monarch RNase A (New England Biolabs) for 1 h at 37 °C. After three PBS washes, the cells were blocked with PBS-0.1%Tween and 10% FBS for 30 min at 37 °C. Next, the BG4 antibody[67] was

incubated 1:500 for 1 h at 37 °C and subsequently incubated with 1:800 mouse anti-FLAG antibody (F1804, Sigma) to BG4 and 1:500 rabbit anti-PARP1 (ab227244, Abcam) for 1 h at 37 °C in PBS-0.1%Tween and 10% FBS. The standard PLA protocol described above was followed to complete sample processing.

## DNA fiber assay

Similar as previously described[17], cells were labeled by sequential incorporation of two different nucleoside analogs, IdU and CldU, into nascent DNA strands for the indicated time and conditions. After nucleoside analogs were incorporated in vivo, the cells were collected, washed, spotted, and lysed on positively charged microscope slides with 7.5 mL spreading buffer for 8 min at room temperature. For experiments with the ssDNA-specific endonuclease S1, after the CldU pulse, cells were treated with CSK100 buffer for 10 min at room temperature, then incubated with S1 nuclease buffer with or without 20 U/mL S1 nuclease (Invitrogen, 18001-016) for 30 min at 37 °C. The cells were then scraped in PBS + 0.1% BSA and centrifuged at 5200xg for 5 min at 4 °C. Cell pellets were resuspended at 1,500 cells/mL and lysed with lysis solution on slides. Individual DNA fibers were released and spread by tilting the slides at 45 degrees. After air-drying, fibers were fixed by 3:1 methanol/acetic acid at room temperature for 3 min. After air-drying again, fibers were rehydrated in PBS, denatured with 2.5 M HCl for 30 min, washed with PBS, and blocked with blocking buffer (PBS + 0.1% Triton X-100 + 3% BSA) for 1 h. Next, slides were incubated for 2.5 h with primary antibodies (IdU, Becton Dickinson 347580 1:100; CldU, Abcam 6326 1:100) diluted in blocking buffer, washed several times in PBS, and then incubated with secondary antibodies (IdU, goat anti-mouse, Alexa Fluor 488 1:200; CldU, goat anti-rat, Alexa Fluor 594 1:200) in blocking buffer for 1 h. After washing and air-drying, slides were mounted with Prolong (Invitrogen P36930). Finally, green and/or red signals (measure at least 100 fibers for each experiment) were visualized by fluorescence microscopy (Axioplan 2 imaging, Zeiss) for the active replication at the single-molecule level.

## Statistical analysis

Statistical differences in DNA fiber assays and immunofluorescence intensity were determined by nonparametric Kruskal-Wallis test followed by Dunn's test for multiple comparisons in non-Gaussian populations. Two group comparisons were determined using two-tailed Mann–Whitney test. Statistical differences in viability assays with small sample sizes were determined by two-way ANOVAs followed by Tukey's test (two-independent variables with multiple comparisons), one-way ANOVAs followed by Dunnett's test (multiple comparisons) or unpaired t-test (two-group comparison, two-tailed, unequal variance). Statistical analysis was performed using GraphPad Prism (v9.2.0).

## Reporting summary

Further information on research design is available in the Nature Portfolio Reporting Summary linked to this article.

# Data availability

We have deposited all original quality westerns and select representative microscopy images in the Figshare Repository (https://doi.org/10.6084/m9.figshare.24280162). All materials associated with this study are available upon request from the correspdoning author. Further information and requests for resources and reagents should be directed to and will be fulfilled by the correspdoning author. Source data are provided as a Source Data file. Source data are provided with this paper.

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

## Acknowledgements

We thank the members of the Cantor laboratory for helpful discussions. We thank Dr. Daniel Durocher for the RPE1 *BRCA1* K/O and isogenic matched WT RPE1 cells, Dr. Keith Caldecott for the *XRCC1* KO, *PARP1* KO,

*XRCC1/PARP1* DKO and matched isogenic WT RPE1 cells, Dr. Christopher Heinen for the HeLa *MSH2* K/O and matched isogenic HeLa WT cells and human endometrial HEC59 and HEC59 + chr2 cell lines, and Dr. Hans Joenje for the FA-J (EUFA30-F) cells. This work was supported by R01 CA254037, R01 CA225018-02, Fanconi Anemia Research Fund, and charitable contributions from Mr. and Mrs. Edward T. Vitone, Jr (S.C.), and NIH K01 AG056554 and an NSF CAREER Award 2143016 (T.D.).

## Author contributions

S.C., T.D., N.M. and K.C. designed the experiments. K.C., N.M., S.L., S.G.M., J.C., M.P. and A.N.K. performed the experiments. K.C., N.M., S.L., and J.C. analyzed the data. S.C., N.M. and K.C. wrote the manuscript. S.C. and T.D. supervised the research.

## Competing interests

The authors declare no competing interests.
