## [Peer Review File · Nature Communications]

FANCI promotes PARP1 activity during DNA replication that is essential in BRCA1 deficient cellsREVIEWER COMMENTS

Reviewer #1 (Remarks to the Author):

The activity of PARP inhibitors (PARPi) in BRCA1/2 deficient cells is frequently attributed to the formation of trapped PARP1 at single-stranded DNA breaks (SSBs). Collision of replication forks with trapped PARP or SSBs may then lead to DSBs that require BRCA1/2-mediated homologous recombination (HR) and/or fork protection to avoid cytotoxicity. Alternatively, the formation of persistent ssDNA gaps has emerged as a reliable predictor of PARPi toxicity, and this has for example been attributed to the activity of PARP1 in S-phase specifically during incomplete lagging strand synthesis. This model is in part supported by the observation that the cells with a deficiency in the FANCD1 helicase show reduced gap formation and sensitivity following PARPi treatment.

In the present manuscript, the authors make use of the unique activity of FANCD1 to elucidate that reduced PARP1 activity in S-phase can limit PARPi-induced toxicity in BRCA1/2 deficient cells. The authors show that not the intrinsic poly(ADP-ribose) activity in response to DNA damage, but S-phase specific activity of PARP1 is affected in FANCD1 deficient cells. Drawing from iPOND and PLA assays, the authors show that both PARP1 and MSH2 are enriched in chromatin in FANCD1 deficient cells. They find that the well known defect in resolving G4 sequences by FANCD1 correlates with limiting PARP1 activity in S-phase, suggesting that PARP1 can be trapped at G4s. Consistently, they show that in the presence of MSH2, G4s are enriched and PARP1 activity and PARPi sensitivity are reduced. Lastly, they show that the absence of FANCD1 is synthetic sick with BRCA1 deficiency, consistent with the importance of PARP1 S-phase activity in BRCA cells.

The findings in this study are sound and verified in multiple cell lines and provide significant new insights into the mechanism of action of PARPi in BRCA deficient cells. Importantly, they show that full PARP1 activity in S-phase is needed for a therapeutic effect and that this activity can be restricted by the sequestration of PARP1 at G4 and possibly other non-canonical DNA structures. The distinction of trapping PARP as SSBs and at G4 structures is an important finding in the understanding of PARP biology. The finding that FANCD1 restricts full PARP1 activity in S-phase furthermore reveals FANCD1 as a potential drug target for BRCA deficient cells. Altogether, these findings warrant publication in Nat Comm after the following issues are addressed.

1. Fig 1b,c. For the clarity of the reader, it would be desirable to specify in the text when PAR levels were determined by WB (Fig 1b) or immunofluorescence (Fig 1C).

2. Figs 2, 4, S2, S3, S4, S5: clonogenic survival assays should be subjected to statistical analysis to establish the significance of the survival assays, which would make all of these figures and especially Fig S4h more convincing.

3. Fig 5. It would help the reader appreciate the significance of this study this figure also included a model highlighting the role of MSH2 in modulating PARP activity.

4. I think the manuscript could benefit from a clearer discussion of two implications of the findings in the discussion – namely the value an assay to predict the levels of PARP activity in S-phase for therapy and the possibility that FANCD1 could be a drug target in BRCA deficient cells.

Reviewer #2 (Remarks to the Author):

This manuscript addresses the role of FANCD1 in regulating the activity of PARP1. The authors show that, in FANCD1-deficient cells, PARP1 is sequestered at G4 secondary DNA structures. This PARP1 sequestration is dependent on MSH2, which interacts with PARP1 at those structures, and is suppressed by the FANCD1-MLH1 complex formation. While many mechanistic details of this process are still missing (such as: what is the function of MSH2 in stabilizing PARP1 at G4, how is MLH1-FANCD1 inhibiting this), I believe that overall this study is important since it provides important insights into PARP1 functions, and allows the authors to gain insights into the mechanism of action of PARP inhibitors. The idea that loss of PARP1 activity in S-phase represents the main determinant of toxicity in genetic backgrounds such as BRCA1 mutations, and that this can be achieved in other ways than PARP1 inhibitor-induced trapping (such as FANCD1 inactivation) could ultimately have clinical implications. Therefore, in my view, the manuscript is worthy of publication in Nature Communications. My comments below reflect mainly the need to provide additional experiments and controls needed to shore up the conclusions drawn, as well as to improve the presentation of the findings.

Specific comments:

1. In general, the manuscript is written in a very brief manner, without carefully explaining the rationale of the various experimental setups used. For example, the use of PARPi in fig 1 is not addressed in the text. The authors need to add an explanation of why this treatment is needed.

2. I am also somewhat confused by the phrasing “PARP1 DNA replication activity”. It is not very descriptive and it may be misleading to the readers as it suggests an activity of PARP1 in DNA replication. Something along the lines of “PARP1 activity during DNA replication” may be more appropriate.

3. The schematic in fig 1f is not well explained. It is unclear based on what data the authors put this together. Is it a pure speculation? Is it based on previously published data?

4. The experiments measuring PAR chain formation upon DNA damage exposure (suppl fig 1) need to be also performed in the presence of PARGi. Otherwise they cannot be compared with the results obtained under normal conditions (fig 1b,e). Without this important control, the authors cannot conclude FANCD1 controls PAR levels only under normal conditions and not in response to DNA damage (since PARGi treatment upon DNA damage exposure may reveal differences between WT and KO cells).

5. The PARP1 chromatin localization in FANCD1 KO cells (fig 1h,i) is shown only in RPE1 cells. Can the results be reproduced in U2OS and 293T cells?

6. Similarly, the MSH2 localization studies (fig 2a,b) were performed only in RPE1 cells with a single FANCD1 KO clone. It is important that the authors confirm the results with their U2OS KO cells.

7. Do other MMR factors show similar localization patterns as shown for MSH2 in fig 2a,b? Investigating MSH3 and MLH1 may be very informative.

8. The authors state that the results in fig 2b suggest that PARP1 and MSH2 are “aberrantly elevated” but in my opinion they cannot draw this conclusion, since the results presented only show an increased interaction between themselves and with PCNA. No data is presented to show that the protein levels overall are elevated.

9. In my opinion, it is important that the authors investigate the proposed “PARP1 sequestration” at G4 structures in FANCD1-deficient cells. PLA assays using the PARP1 and BG4 antibodies would be a direct approach to investigate this. A more indirect approach could be to induce formation of G4 structures (for example by treatment with G4-stabilizing compounds) and investigate the impact on PARP1.

10. The S1 nuclease assay in fig 1e seems to show increased replication fork speed upon knockdown of MSH2 in FANCD1 KA mutant cells treated with olaparib. The authors do not comment on this. Does this perhaps reflect increased PRIMPOL-mediated repriming? PRIMPOL knockdown studies should be used to address this.

11. In any case, to shore up this aspect of the study, a separate experimental setup for the S1 assay should also be used to confirm the findings reported in fig 1e: treatment with olaparib only during incubation with the second analog (CldU) and measuring the ratio between the CldU and IdU tracts.

12. Finally, the authors should also include in fig 4e the important control of MSH2 knockdown in WT cells (not only in the mutant FANCI cells shown).

13. The results presented in fig 5e suggesting a synthetic lethality between BRCA1 and FANCI also need to be shored up using separate approaches. At the very least, the authors should show this relationship using si/shRNA (knocking down FANCI in BRCA1 KO cells and the other way around).

14. Is there a similar synthetic lethality interaction between FANCI and BRCA2? If not, why is this specific for BRCA1?

Reviewer #3 (Remarks to the Author):

In their study “FANCI promotes PARP1 DNA replication activity that is essential in BRCA1 deficient cells” Cong and MacGilvary et al. present evidence that PARP’s DNA replication dependent activity relies upon FANCI and MSH2, and provide fairly clear, largely immunofluorescent based data that demonstrate a differential PARP response based on the presence or absence of FANCI. I am supportive of this work in general, and have a variety of recommendations to consolidate the study.

In general, reader confidence in some of the primary conclusions would benefit from experiments that measure PARP localization that are not immunofluorescence based. Strongly recommend performing a ChIP on common G4 areas to see if PARP sequestering was altered in cells mutated for MSH2, FANCI or the combo. While the authors have performed a PLA assay, one experiment that would help support their model would be a PLA of G4 quadruplexes (using their antibody) and PARP under each of mutant conditions.

The reintroduction of chromosome 2 as a way to reintroduce MSH2 expression is odd – why not use just the MSH2 cDNA from a plasmid? Barring some technical reason why this cannot be done, I would rather see ectopic MSH2 expression from a plasmid, or siRNA/gene knockout, as a way to manipulate MSH2 expression as it seems like a much cleaner system than what has been used. If using chromosome 2 is essential, is there a way to mutate or delete MSH2 from this as a negative control, to assess the very possible impact of every other gene on that chromosome eliciting unknown effects? Also, it would seem

that the outcomes of these experiments disagree with the overall model where authors see less chromatin bound PARP1 with restoration of MSH2, but also attenuation of PARP1 activity.

EdU has been demonstrated to introduce cross-links and, although effective at indicating S-phase, is not without its own effects on DNA replication and DNA damage responses. This has to be considered, as the act of using EdU may alter outcomes in unexpected ways. I say this from our own teams (and that of others) experience using EdU – it really can confound what one is looking at in terms of S-Phase DNA damage responses, especially when PARP inhibitors are in play. Replicating key findings with a non-invasive marker such as a cdk-cyclin marker might help eliminate the possibility of confounding effects.

There appears to be the faintest band for FANCI in U2OS and RPE1? Has loss of expression been confirmed by genetic sequencing, and qPCR to validate knockout of cell lines? It was odd that data using RPE-1 cells data was typically consigned to the supplementary data, whereas this would be far better presented in the main manuscript as generally a normal-derived cell system is preferable to a cancer-derived line for work of this type. This would also help condense the manuscript.

The clonogenic assays often have huge colonies that would be hard to distinguish. How is the team quantifying their clonogenic assays? The size variations of the colonies is concerning for quantification as well, and details on how this is being controlled for is needed. For example, in suppl. fig 5b many of the colonies seem totally unquantifiable, as they've turned into a lawn in many regions of the plate.

FANCI-KO cells exhibit increased amounts of chromatin bound PARP1 but reduced PARylation, and the authors rule out that the issue could be due to inability to effectively load PARP1 onto DNA. Why?

Within the introduction, first paragraph, it's worth mentioning the specifics of PARP1's broader roles, as the authors simply mention SSB repair and replication forks. At minimum, the characterized roles in other repair pathways notably NER and alt-NHEJ should be touched upon. The acronym HBOC is used rarely, not necessary to abbreviate. Similarly in Paragraph 2, Fork protection (FP) does not need acronym, also only used once throughout text. Generally, recommend removing all abbreviations unless used very frequently. The sentence "FANCI deficient cells have low PARPi induced sensitivity..." is unclear, and I am unsure whether authors are stating that FANCI deficient cells are sensitive to PARPi, or the combination of PARP inhibition and loss of FANCI makes these cells susceptible to chemical insults. Please reword for clarity. In paragraph 3 the statement "... restored by depletion of the MMR protein, MSH2 suggesting ... " comma should come after MSH2 to read as "restored by depletion of the MMR protein MSH2, suggesting ... "

Other comments:

- Figure 1b – Need quantification of western blot signal in reference to total protein loaded (as determined by stain free gels (gold standard) or to loading controls (not as robust but ok), done in triplicate. This would further support claim that PARylation activity is reduced in FANCI knockout cells.
 - Figure 1c – Would appreciate if the authors indicated whether they are reporting mean intensity of integrated intensity of PAR signal in nuclei.
 - Figure 1h, 2e, 2f, 3b, 3c, 3d, 4f – Some representative images of these experiments would be appreciated and beneficial. For many of these, stats are also missing altogether or under-defined. Greater overall attention to stats on all data is encouraged.
 - Figure 2g – a positive control for formation of g-quadruplexes is needed to validate outcomes, though there isn't a huge issue with controversy about this antibody compared to some other DNA antibodies. Could use telomestatin or other ligand to stabilize G-quadruplexes as positive control to provide some sort of validation.
 - Figure 2h – This is very interesting! Given the bimodal distribution for PAR intensity in the gene-edited cells for DMSO treatments, would like the authors to provide sequence confirmation for both alleles to confirm that they were both edited in these cell lines, as that could be a reason they see this distribution.
 - Figure 3a – In addition to chromatin bound PARP1, are PARP1 expression levels unaffected? Loading of samples seems uneven given H2B band intensities.
 - Figure 3c – Are the cells used here synchronized for this experiment, otherwise that could be a confounding factor.
 - Figure 3d – The norm for RPA analysis is foci/object counting, not just use of intensity measurements. Please score the number of RPA foci in EdU/S-phase marker positive cells. These can be corrected for intensity, and nuclear volume as well.
 - Figure 3e – Not the most clear blot. Recommend quantifying the blot to further support the claim.
- Figure 4a – Supplemental data S4a through S4c should really take the place of 4a. It's a lot more convincing data.
- Figure 4b – Cannot easily see how Figure S4d is relevant to this? Looks like missing positive error bar on k141/142a with siMSH2. Difference in means appears to be minuscule. Need to know what these lines and error bars represent (are they mean, geometric, median?). Would appreciate a positive control for G4 formation using small molecule inhibitor in this experiment and what effect MSH2 knockdown has on it. Also, are these EdU positive cells only, or all cells in general? That may explain why there is such a large population with next to no foci.
 - Figure 4c – There is a clear bimodal with shRNA b – why? There is also much clearer phenotypes in the FA-J cells (supplemental). Need representative images and also how quantifying to obtain such low intensities on y-axis.
 - Figure 4d – Would appreciate more detailed explanation of what is being reported on y-axis of graph. Also, the authors report that restoration of MSH2 by chromosome 2 reintroduction leads to reduced

chromatin bound PARP1, whereas they are seeing the opposite phenotype in the RPE-1 cells – why the apparent contradiction?

- Figure 5a – The caption for this figure lacks key details. From the figure, it appears the authors are suggesting that G4 sequestration is distinct from PARP trapping from inhibitors, and this should be reflected in the caption.
- Figure 5e – Seems to be missing clonogenic viability in reference to wildtype?

In response to Reviewer #1:

The activity of PARP inhibitors (PARPi) in BRCA1/2 deficient cells is frequently attributed to the formation of trapped PARP1 at single-stranded DNA breaks (SSBs). Collision of replication forks with trapped PARP or SSBs may then lead to DSBs that require BRCA1/2-mediated homologous recombination (HR) and/or fork protection to avoid cytotoxicity. Alternatively, the formation of persistent ssDNA gaps has emerged as a reliable predictor of PARPi toxicity, and this has for example been attributed to the activity of PARP1 in S-phase specifically during incomplete lagging strand synthesis. This model is in part supported by the observation that the cells with a deficiency in the FANCD1 helicase show reduced gap formation and sensitivity following PARPi treatment.

In the present manuscript, the authors make use of the unique activity of FANCD1 to elucidate that reduced PARP1 activity in S-phase can limit PARPi-induced toxicity in BRCA1/2 deficient cells. The authors show that not the intrinsic poly(ADP-ribose)ylation activity in response to DNA damage, but S-phase specific activity of PARP1 is affected in FANCD1 deficient cells. Drawing from iPOND and PLA assays, the authors show that both PARP1 and MSH2 are enriched in chromatin in FANCD1 deficient cells. They find that the well known defect in resolving G4 sequences by FANCD1 correlates with limiting PARP1 activity in S-phase, suggesting that PARP1 can be trapped at G4s. Consistently, they show that in the presence of MSH2, G4s are enriched and PARP1 activity and PARPi sensitivity are reduced. Lastly, they show that the absence of FANCD1 is synthetic sick with BRCA1 deficiency, consistent with the importance of PARP1 S-phase activity in BRCA cells.

The findings in this study are sound and verified in multiple cell lines and provide significant new insights into the mechanism of action of PARPi in BRCA deficient cells. Importantly, they show that full PARP1 activity in S-phase is needed for a therapeutic effect and that this activity can be restricted by the sequestration of PARP1 at G4 and possibly other non-canonical DNA structures. The distinction of trapping PARP as SSBs and at G4 structures is an important finding in the understanding of PARP biology. The finding that FANCD1 restricts full PARP1 activity in S-phase furthermore reveals FANCD1 as a

potential drug target for BRCA deficient cells. Altogether, these findings warrant publication in Nat Comm after the following issues are addressed.

Thank you very much for the favorable evaluation of our study.

Comments:

1. Fig 1b,c. For the clarity of the reader, it would be desirable to specify in the text when PAR levels were determined by WB (Fig 1b) or immunofluorescence (Fig 1C).

In response, we now specify in the text how PAR levels were determined for revised Fig 1b and e.

2. Figs 2, 4, S2, S3, S4, S5: clonogenic survival assays should be subjected to statistical analysis to establish the significance of the survival assays, which would make all of these figures and especially Fig S4h more convincing.

We agree, and therefore re-analyzed these survival assays and also improved the statistics in figure 5 by either a two-way ANOVA followed by Tukey's test (two-independent variables with multiple comparisons), one-way ANOVAs followed by Dunnett's test (multiple-group comparison), unpaired t-test (two-group comparison)^{1,2} and describe in the revised methods section.

3. Fig 5. It would help the reader appreciate the significance of this study this figure also included a model highlighting the role of MSH2 in modulating PARP activity.

Thank you for the suggestion, we have modified this figure to include MSH2 and include an additional model for regulation in WT cells (Figure S5i).

4. I think the manuscript could benefit from a clearer discussion of two implications of the findings in the discussion – namely the value an assay to predict the levels of PARP activity in S-phase for therapy and the possibility that FANCD1 could be a drug target in BRCA deficient cells. We thank you for your comments and have added some additional points in the discussion. We agree as proposed by others that PAR levels could serve as a biomarker of PARPi efficacy³, but temper this point given that PAR levels in S phase could be altered by an unappreciated source of DNA damage. We also include the potential therapeutic benefit of targeting FANCD1 in BRCA1 mutant cells.

In response to Reviewer #2:

This manuscript addresses the role of FANCD1 in regulating the activity of PARP1. The authors show that, in FANCD1-deficient cells, PARP1 is sequestered at G4 secondary DNA structures. This PARP1 sequestration is dependent on MSH2, which interacts with PARP1 at those structures, and is suppressed by the FANCD1-MLH1 complex formation. While many mechanistic details of this process are still missing (such as: what is the function of MSH2 in stabilizing PARP1 at G4, how is MLH1-FANCD1 inhibiting this), I believe that overall this study is important since it provides important insights into PARP1 functions, and allows the authors to gain insights into the mechanism of action of PARP inhibitors. The idea that loss of PARP1 activity in S-phase represents the main determinant of toxicity in genetic backgrounds such as BRCA1 mutations, and that this can be achieved in other ways that PARP1 inhibitor-induced trapping (such as FANCD1 inactivation) could ultimately have clinical implications. Therefore, in my view, the manuscript is worthy of publication in Nature Communications. My comments below reflect mainly the need to provide additional experiments and controls needed to shore up the conclusions drawn, as well as to improve the presentation of the findings.

We appreciate your encouraging comments on our work.

Specific comments:

1. In general, the manuscript is written in a very brief manner, without carefully explaining the rationale of the various experimental setups used. For example, the use of PARGi in fig 1 is not addressed in the text. The authors need to add an explanation of why this treatment is needed. We have redrafted the manuscript to enhance details and experimental setups including an explanation for employing PARGi. Additionally, to maintain consistency, we now describe the cell line control cells as WT throughout the revised manuscript.

2. I am also somewhat confused by the phrasing “PARP1 DNA replication activity”. It is not very descriptive and it may be misleading to the readers as it suggests an activity of PARP1 in DNA replication. Something along the lines of “PARP1 activity during DNA replication” may be more appropriate. We appreciate the advice and have adopted the improved language in the title and manuscript.

3. The schematic in fig 1f is not well explained. It is unclear based on what data the authors put this together. Is it a pure speculation? Is it based on previously published data? Sorry for the confusion. This model is speculation, and we clarify in the text that we sought to depict two possible explanations for the low PAR in FANCD1 deficient cells; PARP1 fails to be loaded or PARP1 fails to be unloaded from in chromatin in S phase cells.

4. The experiments measuring PAR chain formation upon DNA damage exposure (suppl fig 1) need to be also performed in the presence of PARGi. Otherwise they cannot be compared with the results obtained under normal conditions (fig 1b,e). Without this important control, the authors cannot conclude FANCD1 controls PAR levels only under normal conditions and not in response to DNA damage (since PARGi treatment upon DNA damage exposure may reveal differences between WT and KO cells). We appreciate the reviewer’s concern and clarify that PARGi is essential to detect PAR in unperturbed S phase⁴, however in response to DNA damage, PARP1 activity is robust and readily detected without PARGi and inclusion of PARGi provides no detectable gain in PAR levels. We have clarified this point in the revised manuscript.

5. The PARP1 chromatin localization in FANCD1 KO cells (fig 1h,i) is shown only in RPE1 cells. Can the results be reproduced in U2OS and 293T cells? In response, in the revised manuscript we include additional data demonstrating that PARP1 (along MSH2) is enriched in the chromatin of FANCD1 KO U2OS and RPE1 cells by immunoblot (Revised Figure 2d and S2d). Furthermore, this enrichment of PARP1/MSH2 occurs in the WT U2OS cells following treatment with the G4-stabilizing compound, PDS (Revised Figure 2d). Furthermore, as requested by reviewer 3, we find an enriched proximity between PARP1 and G4s in FANCD1 KO U2OS cells (Revised Figure 2c). Collectively, these results along with our prior results showing enriched PARP1 in the chromatin and replisome of FANCD1 null 293T cells⁵ provide strong support for the role of FANCD1 and G4s in regulating PARP1 chromatin association.

6. Similarly, the MSH2 localization studies (fig 2a,b) were performed only in RPE1 cells with a single FANCD1 KO clone. It is important that the authors confirm the results with their U2OS KO cells. Following this suggestion, please refer to point 5 above.

7. Do other MMR factors show similar localization patterns as shown for MSH2 in fig 2a,b? Investigating MSH3 and MLH1 may be very informative. This is an important point, and we refer to our prior iPOND experiments in which we identified MSH2 and MSH6 as enriched in the replisome of FANCD1 KO 293T cells⁵. While we previously found that MSH2 or MSH6 depletion elevated cisplatin resistance in cells lacking the FANCD1-MLH1 interaction, we did not detect a similar rescue with MSH3 depletion⁶. Overall, these findings suggest that FANCD1 through its MLH1 interaction regulates the MSH2-MSH6 complex.

8. The authors state that the results in fig 2b suggest that PARP1 and MSH2 are “aberrantly elevated” but in my opinion they cannot draw this conclusion, since the results presented only show an increased

interaction between themselves and with PCNA. No data is presented to show that the protein levels overall are elevated. We agree and have modified our language to clarify that FANCD1 loss leads to greater proximity between PARP1 and MSH2.

9. In my opinion, it is important that the authors investigate the proposed “PARP1 sequestration” at G4 structures in FANCD1-deficient cells. PLA assays using the PARP1 and BG4 antibodies would be a direct approach to investigate this. A more indirect approach could be to induce formation of G4 structures (for example by treatment with G4-stabilizing compounds) and investigate the impact on PARP1. This is an important suggestion that is also raised by reviewer 3. We now report that as compared to WT, a PARP1-G4 PLA signal is greater in FANCD1 KO U2OS cells suggesting that PARP1 and G4s are in greater proximity (revised Figure 2c) and consistent with other reports^{7,8}. Additionally, as described above, point 5, we report that PDS enriches both MSH2 and PARP1 in chromatin as analyzed by immunoblot (revised Figure 2d).

10. The S1 nuclease assay in fig 1e seems to show increased replication fork speed upon knockdown of MSH2 in FANCD1 KA mutant cells treated with olaparib. The authors do not comment on this. Does this perhaps reflect increased PRIMPOL-mediated repriming? PRIMPOL knockdown studies should be used to address this. The reviewer is correct in that PARPi appears to accelerate replication as originally described⁹ and further demonstrated upon targeting other lagging strand factors including FEN1, LIG1, and PCNA reviewed in¹⁰. We uncovered that this acceleration manifests in discontinuous replication¹¹ consistent with disrupted lagging strand synthesis following PARPi⁴. We have included greater commentary on these points in the revised manuscript. The relationship between lagging strand defects and PRIMPOL repriming following PARPi remain to be fully understood and is a very active area of continued research, thus we believe it is beyond the scope of this work.

11. In any case, to shore up this aspect of the study, a separate experimental setup for the S1 assay should also be used to confirm the findings reported in fig 1e: treatment with olaparib only during incubation with the second analog (CldU) and measuring the ratio between the CldU and IdU tracts. With respect to assessing gaps in DNA fibers as in Fig. 4e, we find that the experimental conditions require Olaparib to be dosed at a level and duration that allows the most robust detection of lengthening. Unlike other reports¹², we do not readily detect lengthening of tracts when PARPi is restricted to 45min during the second analogue. Instead, our experimental setup is more consistent with other reports^{1,9,13}.

12. Finally, the authors should also include in fig 4e the important control of MSH2 knockdown in WT cells (not only in the mutant FANCD1 cells shown). In response, we include data showing that MSH2 deficiency in two distinct cell lines confers greater PARP1 activity and accordingly PARPi is more effective. Moreover, EdU incorporation and RPA loading are elevated, consistent with ssDNA gaps in replicating cells. Thus, it is expected that MSH2 depletion in WT cells will also increase fork speed and gaps and therefore it is unclear how including these results will further the conclusions of the manuscript. We clarify here and in the revised text that MSH2 and FANCD1 oppose each other in a manner regulated by the FANCD1-MLH1 interaction such that loss of either without the other has little impact whereas the depletion of MSH2 restores FANCD1-MLH1 mutant cells to resemble WT cells.

13. The results presented in fig 5e suggesting a synthetic lethality between BRCA1 and FANCD1 also need to be shored up using separate approaches. At the very least, the authors should show this relationship using si/shRNA (knocking down FANCD1 in BRCA1 KO cells and the other way around). In response, we have performed additional FANCD1 and BRCA1 deficiency studies with siRNA reagents as suggested and find that they validate our original DKO model RPE cell system (Revised Figure S5h). Furthermore, we more clearly describe that another group found the same loss of fitness genetic interaction between FANCD1 and BRCA1¹⁴.

14. Is there a similar synthetic lethality interaction between FANCD1 and BRCA2? If not, why is this specific for BRCA1? The reviewer asks an important question that we attempted to address in two

distinct BRCA2 deficient cell backgrounds, the BRCA2 mutant ovarian cancer cell line PEO1 and its matched BRCA2 reversion cell line C4-2; and via depletion of BRCA2 in FANCI KO vs WT RPE1 cells. FANCI depletion in the PEO1 cells is tolerated similar to non-silencing control. Similarly, BRCA2 depletion in the FANCI KO RPE1 cells is well tolerated. Thus, it appears that FANCI and BRCA2 deficiency is distinct from FANCI and BRCA1 deficiency. However, we prefer not to include this preliminary data for two reasons. For one, we do not have a mechanistic understanding for this distinction in part because we have yet to explore the role of BRCA2 in lagging strand synthesis. Thus, it would take additional experiments to understand how BRCA1 and BRCA2 differ with respect to FANCI dependence that we feel warrants investigation in a subsequent manuscript. Additionally, we had an unexpected finding, that FANCI depletion in the C4-2 cell line is toxic. Perhaps we have identified that FANCI is a unique vulnerability in this reversion cell line?

In response to Reviewer #3

In their study “FANCI promotes PARP1 DNA replication activity that is essential in BRCA1 deficient cells” Cong and MacGilvary et al. present evidence that PARP’s DNA replication dependent activity relies upon FANCI and MSH2, and provide fairly clear, largely immunofluorescent based data that demonstrate a differential PARP response based on the presence or absence of FANCI. I am supportive of this work in general, and have a variety of recommendations to consolidate the study.

Thank you for your support and recommendations for our work.

In general, reader confidence in some of the primary conclusions would benefit from experiments that measure PARP localization that are not immunofluorescence based. Strongly recommend performing a ChIP on common G4 areas to see if PARP sequestering was altered in cells mutated for MSH2, FANCI or the combo. While the authors have performed a PLA assay, one experiment that would help support their model would be a PLA of G4 quadruplexes (using their antibody) and PARP under each of mutant conditions. As described above, we agree this is an important suggestion. We favored the PLA approach to gain information on the proximity of PARP1 to G4s over ChIP that would be limited to the residence of PARP1 on G4-like sequences. Importantly, we now report that as compared to WT, a PARP1-G4 PLA is greater in FANCI KO U2OS cells (revised Fig 2c). Additionally, as described above, reviewer 2 point 5, we also find that PDS enriches both MSH2 and PARP1 in chromatin as detected by immunoblot (revised Fig 2d). Unfortunately, our attempts to perform the PARP-G4 PLA in RPE1 cells have been problematic. While we can detect a reproducible signal in the helicase dead and MLH1 binding mutant backgrounds, we do not observe a signal in the FANCI KO cells that is above WT control. Conceivably, this reflects U2OS cells having greater S phase G4s and thus detection is achieved whereas in the RPE1 cells, only under conditions of the helicase inactive mutant can a PLA be detected consistent with this genetic background also having higher G4s (revised Figure 2c). In summary, we now include our U2OS data that provides additional support for PARP1-G4 proximity as reported by others^{7,8}.

The reintroduction of chromosome 2 as a way to reintroduce MSH2 expression is odd – why not use just the MSH2 cDNA from a plasmid? Barring some technical reason why this cannot be done, I would rather see ectopic MSH2 expression from a plasmid, or siRNA/gene knockout, as a way to manipulate MSH2 expression as it seems like a much cleaner system than what has been used. If using chromosome 2 is essential, is there a way to mutate or delete MSH2 from this as a negative control, to assess the very possible impact of every other gene on that chromosome eliciting unknown effects? Also, it would seem that the outcomes of these experiments disagree with the overall model where authors see less chromatin bound PARP1 with restoration of MSH2, but also attenuation of PARP1 activity. The reviewer brings up an important point that we now clarify in the revised manuscript. Historically, introduction of MMR genes into MMR deficient cells has not been successful for reasons that are not understood but could stem from MMR restricted PARP1 activity. Thus, to re-establish MSH2 in the HEC59 endometrial

line an entire chromosome is introduced¹⁵. Given the potential limitations of this system, we agree with the reviewer and had employed HeLa cells in which MSH2 was deleted vs a WT control to verify the PARP1 biology. We also clarify that both the HEC59 and HeLa systems demonstrate the same relationship, reduced MSH2 confers greater PARP1 activity, PARPi sensitivity and reduced G4s.

EdU has been demonstrated to introduce cross-links and, although effective at indicating S-phase, is not without its own effects on DNA replication and DNA damage responses. This has to be considered, as the act of using EdU may alter outcomes in unexpected ways. I say this from our own teams (and that of others) experience using EdU – it really can confound what one is looking at in terms of S-Phase DNA damage responses, especially when PARP inhibitors are in play. Replicating key findings with a non-invasive marker such as a cdk-cyclin marker might help eliminate the possibility of confounding effects. We appreciate the reviewer's concern and suggest that while with limitations it provides an assessment of replication in a manner that can be compared across this study and others. We also make comparisons by PLA and immunoblot without employing analogues.

There appears to be the faintest band for FANCD1 in U2OS and RPE1? Has loss of expression been confirmed by genetic sequencing, and qPCR to validate knockout of cell lines? It was odd that data using RPE-1 cells data was typically consigned to the supplementary data, whereas this would be far better presented in the main manuscript as generally a normal-derived cell system is preferable to a cancer-derived line for work of this type. This would also help condense the manuscript. Thanks for this comment. We have clarified in the revised manuscript methods that the FANCD1 null cell lines employed were sequenced and verified^{5,11}. Additionally, to validate the combined deletion in the revised manuscript we include an immunoblot confirming FANCD1 and BRCA1 dual deficiency (Revised Figure S5g). For Figure 1, the goal was to validate the prior finding in RPE1 cells that FANCD1 loss reduces S phase PAR¹¹, thus we prioritize the cancer cell line systems. In the revised manuscript, we have improved the presentation of this prior data to clarify our goals.

The clonogenic assays often have huge colonies that would be hard to distinguish. How is the team quantifying their clonogenic assays? The size variations of the colonies is concerning for quantification as well, and details on how this is being controlled for is needed. For example, in suppl. fig 5b many of the colonies seem totally unquantifiable, as they've turned into a lawn in many regions of the plate. We appreciate this concern and point out that the S5b figure mentioned was included to serve more as a comparison across genetic backgrounds to make a point about basic fitness. For example, the data in S5b, show that XRCC1 and PARP1 loss is not toxic which is in direct contrast to BRCA1 and PARP1 loss which is toxic. Otherwise, quantification in RPE1 is readily observable and our protocol is now included in revised manuscript methods¹⁶ and we have improved the language to reflect this data.

FANCD1-KO cells exhibit increased amounts of chromatin bound PARP1 but reduced PARylation, and the authors rule out that the issue could be due to inability to effectively load PARP1 onto DNA. Why? We clarify that we sought to determine if the low S phase PAR in FANCD1 KO cells stemmed from a defect in PARP1 loading or unloading. We conclude that PARP1 chromatin loading is not the problem because FANCD1 null cells display elevated PARP1 chromatin loading. We hope our improved explanation of the model in Fig 1f provides greater clarity.

Within the introduction, first paragraph, it's worth mentioning the specifics of PARP1's broader roles, as the authors simply mention SSB repair and replication forks. At minimum, the characterized roles in other repair pathways notably NER and alt-NHEJ should be touched upon. The acronym HBOC is used rarely, not necessary to abbreviate. Similarly in Paragraph 2, Fork protection (FP) does not need acronym, also only used once throughout text. Generally, recommend removing all abbreviations unless used very frequently. The sentence "FANCD1 deficient cells have low PARPi induced sensitivity..." is unclear, and I am unsure whether authors are stating that FANCD1 deficient cells are sensitive to PARPi, or the combination of PARP inhibition and loss of FANCD1 makes these cells susceptible to chemical insults. Please reword for clarity. In paragraph 3 the statement "... restored by depletion of the MMR protein,

MSH2 suggesting ... “ comma should come after MSH2 to read as “restored by depletion of the MMR protein MSH2, suggesting ... “ We agree with the reviewer’s assessment and have made several adjustments to reduce abbreviations and improve clarity.

Other comments:

- Figure 1b – Need quantification of western blot signal in reference to total protein loaded (as determined by stain free gels (gold standard) or to loading controls (not as robust but ok), done in triplicate. This would further support claim that PARylation activity is reduced in FANCD1 knockout cells. We have performed the requested quantification of three independent experiments and have added this data below the blot in (revised Figure 1b) that includes three independent experiments showing the average relative PAR as normalized to β -actin. Overall, we find a decrease of total PAR in FANCD1 null cells. However, by comparison, there is a more substantial S phase PAR reduction in FANCD1 null cells as analyzed by IF that can select EdU positive cells. This distinction provides further evidence that FANCD1 functions to activate PARP1 in S phase.
- Figure 1c – Would appreciate if the authors indicated whether they are reporting mean intensity of integrated intensity of PAR signal in nuclei. We clarify in the revised manuscript that this depicts the mean PAR intensity per nuclei (revised Fig. 1e)
- Figure 1h, 2e, 2f, 3b, 3c, 3d, 4f – Some representative images of these experiments would be appreciated and beneficial. For many of these, stats are also missing altogether or under-defined. Greater overall attention to stats on all data is encouraged. Thank you, we have now added the requested images and added the appropriate stats in related revised figures.
- Figure 2g – a positive control for formation of g-quadruplexes is needed to validate outcomes, though there isn’t a huge issue with controversy about this antibody compared to some other DNA antibodies. Could use telomestatin or other ligand to stabilize G-quadruplexes as positive control to provide some sort of validation. As requested, we now include that PDS induces G4s as measured by IF in revised Figure S2b and c.
- Figure 2h – This is very interesting! Given the bimodal distribution for PAR intensity in the gene-edited cells for DMSO treatments, would like the authors to provide sequence confirmation for both alleles to confirm that they were both edited in these cell lines, as that could be a reason they see this distribution. We hesitate to over interpret these distributions in revised Fig 2j for two reasons. For one, these are sequenced and are bi-allelic mutations. Second, when we repeat these studies in the FANCD1 null RPE1 by re-expression of mutants (Figure S2n), we do not detect this distribution. While we could speculate that the bimodal effects are related to different levels of replication in the mutant lines, on closer examination, we note that FANCD1 KO cells have a similar but more modest distribution.
- Figure 3a – In addition to chromatin bound PARP1, are PARP1 expression levels unaffected? Loading of samples seems uneven given H2B band intensities. Thanks for the question. We do not see a change in PARP1 levels in a whole cell extract. This information and accompanying blot are now included in the revised manuscript Figure 3a.
- Figure 3c – Are the cells used here synchronized for this experiment, otherwise that could be a confounding factor. We clarify that the cells are not synchronized, and the data provide a reproducible snapshot that overall replication is distinct between the two cell lines.
- Figure 3d – The norm for RPA analysis is foci/object counting, not just use of intensity measurements. Please score the number of RPA foci in EdU/S-phase marker positive cells. These can be corrected for intensity, and nuclear volume as well. We have followed the analysis procedure that provides an overview of RPA occupancy on ssDNA¹⁷⁻¹⁹ that does not require robust foci for detection.

- Figure 3e – Not the most clear blot. Recommend quantifying the blot to further support the claim. We have included quantification for this blot in triplicate in the revised Figure 3e.

Figure 4a – Supplemental data S4a through S4c should really take the place of 4a. It's a lot more convincing data. We hesitate to bring this supplemental data into the main figure because it serves primarily to verify the functional loss of MSH2 which we previously found restores replication in the mutant cells⁶.

- Figure 4b – Cannot easily see how Figure S4d is relevant to this? Looks like missing positive error bar on k141/142a with siMSH2. Difference in means appears to be minuscule. Need to know what these lines and error bars represent (are they mean, geomean, median?). Would appreciate a positive control for G4 formation using small molecule inhibitor in this experiment and what effect MSH2 knockdown has on it. Also, are these EdU positive cells only, or all cells in general? That may explain why there is such a large population with next to no foci.

We clarify that Figure S4d is relevant because we utilized siRNA in Figure 4b, and this shows depletion was achieved. With respect to the error bar, we are reporting the median \$\pm\$ interquartile range and in this case the positive interquartile range lies at the median resulting in the absence of an error bar. These are total population of cells. We also appreciate the comment as to using a G4 positive control which is now included as requested in Figure S2b and c.

- Figure 4c – There is a clear bimodal with shRNA b – why? There is also much clearer phenotypes in the FA-J cells (supplemental). Need representative images and also how quantifying to obtain such low intensities on y-axis. In response, we do not have a clear explanation of this bimodal other than these represent raw values from multiple independent experiments. We now include representative images in revised Figure S4f.

- Figure 4d – Would appreciate more detailed explanation of what is being reported on y-axis of graph. Also, the authors report that restoration of MSH2 by chromosome 2 reintroduction leads to reduced chromatin bound PARP1, whereas they are seeing the opposite phenotype in the RPE-1 cells – why the apparent contradiction? With respect to Figure 4d, we clarify that the value represents the mean chromatin bound PARP1 intensity. We have also fixed the issue of part of the Y axis being hidden to clarify that this represents mean chromatin bound PARP1 in EdU+ cells. We agree that the WB is limited, and that is why we have addressed the activity of PARP1 in several assays. In both the HeLa and HEC cell lines, MSH2 proficiency consistently shows higher PARP1 chromatin binding, reduced PAR, and accordingly reduced ability to sensitize with a PARPi that requires an active PARP1 enzyme. We have revised this section to improve clarity.

- Figure 5a – The caption for this figure lacks key details. From the figure, it appears the authors are suggesting that G4 sequestration is distinct from PARP trapping from inhibitors, and this should be reflected in the caption. We believe that the caption includes the essential information to understand the figure. We have modified Fig 5a to the following, “Is PARP1 trapping distinct from G4 sequestering?”

- Figure 5e – Seems to be missing clonogenic viability in reference to wildtype?

We thank the reviewer for this suggestion and have added the WT in revised Figure 5e.

Thank you again for reviewing our manuscript. We believe that we have addressed the points raised in full and hope that you will find the revised version suitable for publication in *Nature Communications*.

Sincerely,

Sharon Cantor, Ph.D.
Professor, Molecular, Cell and Cancer Biology
Gladys Smith Martin Chair in Oncology
UMass Chan Medical School

- 1 Thakar, T. *et al.* Lagging strand gap suppression connects BRCA-mediated fork protection to nucleosome assembly through PCNA-dependent CAF-1 recycling. *Nature communications* **13**, 5323, doi:10.1038/s41467-022-33028-y (2022).
- 2 Thakar, T. *et al.* Ubiquitinated-PCNA protects replication forks from DNA2-mediated degradation by regulating Okazaki fragment maturation and chromatin assembly. *Nature communications* **11**, 2147, doi:10.1038/s41467-020-16096-w (2020).
- 3 Cleary, J. M., Aguirre, A. J., Shapiro, G. I. & D'Andrea, A. D. Biomarker-Guided Development of DNA Repair Inhibitors. *Mol Cell* **78**, 1070-1085, doi:10.1016/j.molcel.2020.04.035 (2020).
- 4 Hanzlikova, H. *et al.* The Importance of Poly(ADP-Ribose) Polymerase as a Sensor of Unligated Okazaki Fragments during DNA Replication. *Mol Cell* **71**, 319-331 e313, doi:10.1016/j.molcel.2018.06.004 (2018).
- 5 Peng, M. *et al.* Opposing Roles of FANCI and HLF1 Protect Forks and Restrain Replication during Stress. *Cell Rep* **24**, 3251-3261, doi:10.1016/j.celrep.2018.08.065 (2018).
- 6 Peng, M. *et al.* The FANCI/MutLalpha interaction is required for correction of the cross-link response in FA-J cells. *Embo J* **26**, 3238-3249 (2007).
- 7 Soldatenkov, V. A., Vetcher, A. A., Duka, T. & Ladame, S. First evidence of a functional interaction between DNA quadruplexes and poly(ADP-ribose) polymerase-1. *ACS Chem Biol* **3**, 214-219, doi:10.1021/cb700234f (2008).
- 8 Laspata, N. *et al.* PARP1 associates with R-loops to promote their resolution and genome stability. *Nucleic Acids Res* **51**, 2215-2237, doi:10.1093/nar/gkad066 (2023).
- 9 Maya-Mendoza, A. *et al.* High speed of fork progression induces DNA replication stress and genomic instability. *Nature* **559**, 279-284, doi:10.1038/s41586-018-0261-5 (2018).
- 10 Cong, K. & Cantor, S. B. Exploiting replication gaps for cancer therapy. *Mol Cell* **82**, 2363-2369, doi:10.1016/j.molcel.2022.04.023 (2022).
- 11 Cong, K. *et al.* Replication gaps are a key determinant of PARP inhibitor synthetic lethality with BRCA deficiency. *Mol Cell*, doi:10.1016/j.molcel.2021.06.011 (2021).
- 12 Vaitsiankova, A. *et al.* PARP inhibition impedes the maturation of nascent DNA strands during DNA replication. *Nat Struct Mol Biol* **29**, 329-338, doi:10.1038/s41594-022-00747-1 (2022).
- 13 Mann, A. *et al.* POLtheta prevents MRE11-NBS1-CtIP-dependent fork breakage in the absence of BRCA2/RAD51 by filling lagging-strand gaps. *Mol Cell* **82**, 4218-4231 e4218, doi:10.1016/j.molcel.2022.09.013 (2022).
- 14 Simpson, D. A., Ling, J., Jing Y., Adamson, B. Mapping the Genetic Interaction Network of PARP inhibitor Response. *BioRxiv*, doi:<https://doi.org/10.1101/2023.08.19.553986> (2023).
- 15 Umar, A. *et al.* Correction of hypermutability, N-methyl-N'-nitro-N-nitrosoguanidine resistance, and defective DNA mismatch repair by introducing chromosome 2 into human tumor cells with mutations in MSH2 and MSH6. *Cancer Res* **57**, 3949-3955 (1997).
- 16 Crowley, L. C., Christensen, M. E. & Waterhouse, N. J. Measuring Survival of Adherent Cells with the Colony-Forming Assay. *Cold Spring Harb Protoc* **2016**, doi:10.1101/pdb.prot087171 (2016).
- 17 Toledo, L. I. *et al.* ATR prohibits replication catastrophe by preventing global exhaustion of RPA. *Cell* **155**, 1088-1103, doi:10.1016/j.cell.2013.10.043 (2013).
- 18 Ercilla, A. *et al.* Physiological Tolerance to ssDNA Enables Strand Uncoupling during DNA Replication. *Cell Rep* **30**, 2416-2429 e2417, doi:10.1016/j.celrep.2020.01.067 (2020).
- 19 Somyajit, K. *et al.* Homology-directed repair protects the replicating genome from metabolic assaults. *Dev Cell* **56**, 461-477 e467, doi:10.1016/j.devcel.2021.01.011 (2021).

REVIEWERS' COMMENTS

Reviewer #1 (Remarks to the Author):

In response to the review, the authors have thoroughly revised the manuscript and included additional experiment including:

- i) using U2OS in addition to RPE1 cells for PARP1, MSH2 localization to G4,
- ii) PLA assays to show colocalization of PARP1 and G4,
- iii) demonstration of increased PARP activity in MSH2 kd cells,
- iv) inclusion of additional data to support synthetic lethality between FANCD1 and BRCA1, and
- v) inclusion of statistical analysis in all figures.

Along with extensive modification of the text to increase the clarity of the presentation of the work, this manuscript is now ready for publication in Nat Commun.

Reviewer #2 (Remarks to the Author):

The authors have satisfactorily addressed the issues I raised in my original review. I appreciate that some of the experiments I suggested are not immediately feasible, and I am satisfied with the authors rebuttal. I believe the revised manuscript represents a very strong addition to the field and should be published in Nature Communications

Reviewer #3 (Remarks to the Author):

In their revised study "FANCD1 promotes PARP1 activity during DNA replication that is essential in BRCA1 deficient cells" Cong and MacGilvary et al. address the reviewers concerns in a generally strong revision of their original work. I have reviewed all aspects of the rebuttal for my and the other reviewers recommendations, and consider this study a predominantly consolidated work.

For efficiency, I will not speak to any earlier concerns that I now consider addressed successfully or adequately rebutted – other than to congratulate the authors on a strong performance. My only outstanding recommendation / concern is that since the authors contend on reporting mean RPA intensity versus focal number, I recommend showing representative images that allow reader to visualize a phenotype that looks pretty dramatic. This may help readers who otherwise are skeptical of that approach, "normalizing" use of the approach among the community further.

In response to Reviewer #1:

In response to the review, the authors have thoroughly revised the manuscript and included additional experiment including:

- i) using U2OS in addition to RPE1 cells for PARP1, MSH2 localization to G4,
- ii) PLA assays to show colocalization of PARP1 and G4,
- iii) demonstration of increased PARP activity in MSH2 kd cells,
- iv) inclusion of additional data to support synthetic lethality between FANCI and BRCA1, and
- v) inclusion of statistical analysis in all figures.

Along with extensive modification of the text to increase the clarity of the presentation of the work, this manuscript is now ready for publication in *Nature Communications*.

Thank you for appreciating the modifications to the manuscript and finding it suitable for publication.

In response to Reviewer #2:

The authors have satisfactorily addressed the issues I raised in my original review. I appreciate that some of the experiments I suggested are not immediately feasible, and I am satisfied with the authors rebuttal. I believe the revised manuscript represents a very strong addition to the field and should be published in *Nature Communications*.

Thank you for finding the modifications satisfactory and suitable for publication.

In response to Reviewer #3

In their revised study “FANCI promotes PARP1 activity during DNA replication that is essential in BRCA1 deficient cells” Cong and MacGilvary et al. address the reviewers concerns in a generally strong revision of their original work. I have reviewed all aspects of the rebuttal for my and the other reviewers recommendations, and consider this study a predominantly consolidated work.

For efficiency, I will not speak to any earlier concerns that I now consider addressed successfully or

adequately rebutted – other than to congratulate the authors on a strong performance. My only outstanding recommendation / concern is that since the authors contend on reporting mean RPA intensity versus focal number, I recommend showing representative images that allow reader to visualize a phenotype that looks pretty dramatic. This may help readers who otherwise are skeptical of that approach, “normalizing” use of the approach among the community further.

[Thank you again for your final comment. In response, we now provide higher resolution images deposited in Figshare \(10.6084/m9.figshare.24280162\)](https://doi.org/10.6084/m9.figshare.24280162) that will allow the full representation of RPA stain to be appreciated.

Again, we thank the reviewers for their time and help to make this a much-improved manuscript suitable for publication in *Nature Communications*.

Sincerely,

Sharon Cantor, Ph.D.
Professor, Molecular, Cell and Cancer Biology
Gladys Smith Martin Chair in Oncology
UMass Chan Medical School